# Regulatory pathways governing murine coronary vessel formation are dysregulated in the injured adult heart

Sophie Payne[1,2], Mala Gunadasa-Rohling[2], Alice Neal[1,2], Andia N. Redpath [2], Jyoti Patel[3], Kira M. Chouliaras[1], Indrika Ratnayaka[1], Nicola Smart [2] & Sarah De Val[1,2]

The survival of ischaemic cardiomyocytes after myocardial infarction (MI) depends on the formation of new blood vessels. However, endogenous neovascularization is inefficient and the regulatory pathways directing coronary vessel growth are not well understood. Here we describe three independent regulatory pathways active in coronary vessels during development through analysis of the expression patterns of differentially regulated endothelial enhancers in the heart. The angiogenic VEGFA-MEF2 regulatory pathway is predominantly active in endocardial-derived vessels, whilst SOXF/RBPJ and BMP-SMAD pathways are seen in sinus venosus-derived arterial and venous coronaries, respectively. Although all developmental pathways contribute to post-MI vessel growth in the neonate, none are active during neovascularization after MI in adult hearts. This was particularly notable for the angiogenic VEGFA-MEF2 pathway, otherwise active in adult hearts and during neoangiogenesis in other adult settings. Our results therefore demonstrate a fundamental divergence between the regulation of coronary vessel growth in healthy and ischemic adult hearts.

---

[1] Ludwig Institute for Cancer Research, Nuffield Department of Medicine, University of Oxford, Oxford, UK. [2] BHF Centre of Regenerative Medicine, Department of Physiology, Anatomy and Genetics, University of Oxford, Oxford, UK. [3] Division of Cardiovascular Medicine, BHF Centre of Research Excellence, University of Oxford, John Radcliffe Hospital, Oxford OX3 9DU, UK. Correspondence and requests for materials should be addressed to N.S. (email: nicola.smart@dpag.ox.ac.uk) or to S.D.V. (email: sarah.deval@dpag.ox.ac.uk)

Impaired blood flow to damaged regions of the heart is a common consequence of myocardial infarction (MI). However, the endogenous neovascular response to ischaemic injury in the heart is insufficient, contributing to loss of cardiac function and chronic heart failure[1–3]. While the re-activation of developmental regulatory programmes is a key aim of cardiac regenerative medicine, the signalling and transcriptional pathways involved in coronary vessel growth are not fully elucidated. Furthermore, the role of embryonic and neonatal pathways in adult coronary vascular regeneration is still unclear.

The endothelial cells (ECs) that form the vascular system are first established by differentiation from mesodermal progenitors (vasculogenesis), after which new vessels form primarily from existing ones (angiogenesis). Endothelial heterogeneity is a fundamental feature of the systemic vasculature: differential gene expression in specific endothelial sub-populations is essential for vasculogenesis and angiogenesis; for the creation of the hierarchically branched vascular system of arteries, veins and lymphatics; and for organ specialisation in response to local signals[4–6]. This structural and functional heterogeneity is matched by extensive regulatory heterogeneity, with independent signalling and transcriptional pathways controlling the growth of different endothelial sub-populations[7–9]. Endothelial heterogeneity extends to the coronary vasculature, which similarly contains clearly differentiated arteries and veins. In addition, ECs within the coronaries have multiple, independent origins[10–13]. During mammalian embryonic development, ECs sprout from the sinus venosus (SV) to contribute to many of the coronary arteries, capillaries and veins on the dorsal and lateral sides of the heart[10,11]. Conversely, the vessels on the ventral aspect and within the ventricular septum are thought to originate predominantly from the endocardium[10,12], and a small number of coronary ECs differentiate from the proepicardium[10,13]. ECs from multiple origins also contribute differentially to the rapid vascular growth seen in the neonatal heart: expansion of coronary vessels derived embryonically from the SV provides many of the vessels in the outer myocardial wall, whereas vessels within the inner myocardial wall originate from endocardium and its coronary derivatives[14,15].

Although coronary vessels form primarily from existing vasculature and are therefore angiogenic by definition, there is also evidence of regulatory heterogeneity within the coronary vasculature. Embryonic endocardial cells form coronary vessels in response to vascular endothelial growth factor A (VEGFA)[12], whereas SV-derived coronary vessels instead require VEGFC and ELA/APJ signalling[10,16]. Neonatal endocardial-derived coronary vessels also reportedly use an alternative pathway to that seen in the embryo, arising 'de novo' during trabecular compaction[15]. However, precisely delineating the signalling and transcriptional cascades involved in coronary vessel growth and behaviour has been challenging: knockout models often suffer systemic vascular abnormalities prior to coronary vessel formation, while defects in one type of coronary vessel can be compensated by other sources[16], confounding phenotypic analysis.

Complex spatial and temporal patterns of gene transcription in mammals are primarily regulated by gene enhancers. These are modular elements containing densely clustered groups of transcription factor binding motifs that work cooperatively to activate and enhance transcription[17]. Our lab and others have recently characterised and analysed a number of enhancers that drive gene expression specifically to discrete sub-populations of ECs, clearly demonstrating important roles for epigenetic modification and transcription factor combinations in achieving distinct patterns of gene expression[7,18–23].

Once the precise upstream factors regulating a discrete endothelial enhancer have been defined, transgenic animal models expressing reporter genes under the control of these enhancers become powerful tools to study the behaviours of different regulatory pathways during vascular growth. These enhancer: reporter constructs are not proxies for the gene they regulate, as most gene loci contain multiple enhancers and a single enhancer is rarely active in the entire domain of the gene it regulates. Instead, these enhancer:reporter constructs provide an easily detected read-out of the activity of each different upstream regulatory pathway. This can provide more information than the expression pattern of a single regulatory pathway component: vascular regulatory factors are often expressed in wider domains than the genes they activate, requiring epigenetic modification and/or specific combinations with other factors for gene activation (e.g.,[7,19]). The use of endogenous enhancers also avoids complications associated with synthetic pathway response elements (e.g., BMP or Notch response elements[24,25]), which contain multimerizing consensus binding motifs for a single transcriptional activator. These therefore lack binding sites both for essential endothelial transcription factors (e.g., ETS motifs[26]) and for co-factors required for context-specific activation. Enhancer:reporter transgenic models also differ from Cre-driven lineage tracing, as they are not influenced by recombination efficiency, and switch off once the enhancer is no longer active.

In this paper, we use multiple enhancer:reporter transgenic mouse models to study the activity of different regulatory cascades within the heart during both physiological and pathological coronary vessel growth. This work identifies three independent regulatory pathways differentially active in the coronary vasculature, and demonstrates a fundamental divergence between the regulatory pathways active in healthy and ischaemic adult hearts.

## Results

**MEF2-driven angiogenic pathway is active in coronary vessels.** Within the systemic vasculature, the formation of new vessels from existing ones occurs primarily through sprouting angiogenesis downstream of a VEGFA-induced, MEF2-regulated transcriptional programme[19,27]. During this process, VEGFA triggers the release of repressive class IIa HDACs bound to MEF2 transcription factors. MEF2 factors are expressed widely throughout the systemic endothelium, but the loss of HDAC binding epigenetically converts MEF2 factors from repressors to activators, allowing them to specifically activate enhancers during sprouting angiogenesis[19,28–31].

Conflicting models of coronary vessel formation have implicated sprouting angiogenesis exclusively to different compartments of the vasculature[10,12,15]. Therefore, we wished to directly establish the role of the VEGFA-MEF2 sprouting angiogenic pathway in coronary vascular growth. As in the systemic vasculature, the MEF2 factors MEF2A, C and D were widely expressed throughout the embryonic and neonatal coronary endothelium (as well as in other cardiac cell types; Supplementary Figs. 1 and 2). However, expression of MEF2 factors alone cannot indicate activity of the epigenetically modified VEGFA-MEF2 pathway. To determine this, we instead examined the cardiac activity of the HLX-3:*LacZ* enhancer:reporter transgene, which is directly activated by the VEGFA-MEF2 pathway in ECs: the HLX-3 enhancer, located 3 kb upstream of the homeobox transcription factor *HLX*, activates gene expression specifically during sprouting angiogenesis in the systemic vasculature via direct and essential MEF2 binding[19] (Fig. 1a). Common to all known enhancers active in any endothelial compartment, the HLX-3 enhancer also binds essential ETS transcription factors[19,32]. Analysis of hearts from mice expressing the HLX-3: *lacZ* transgene demonstrated enhancer activity in coronary vessels starting at E11.5 and continuing throughout embryonic

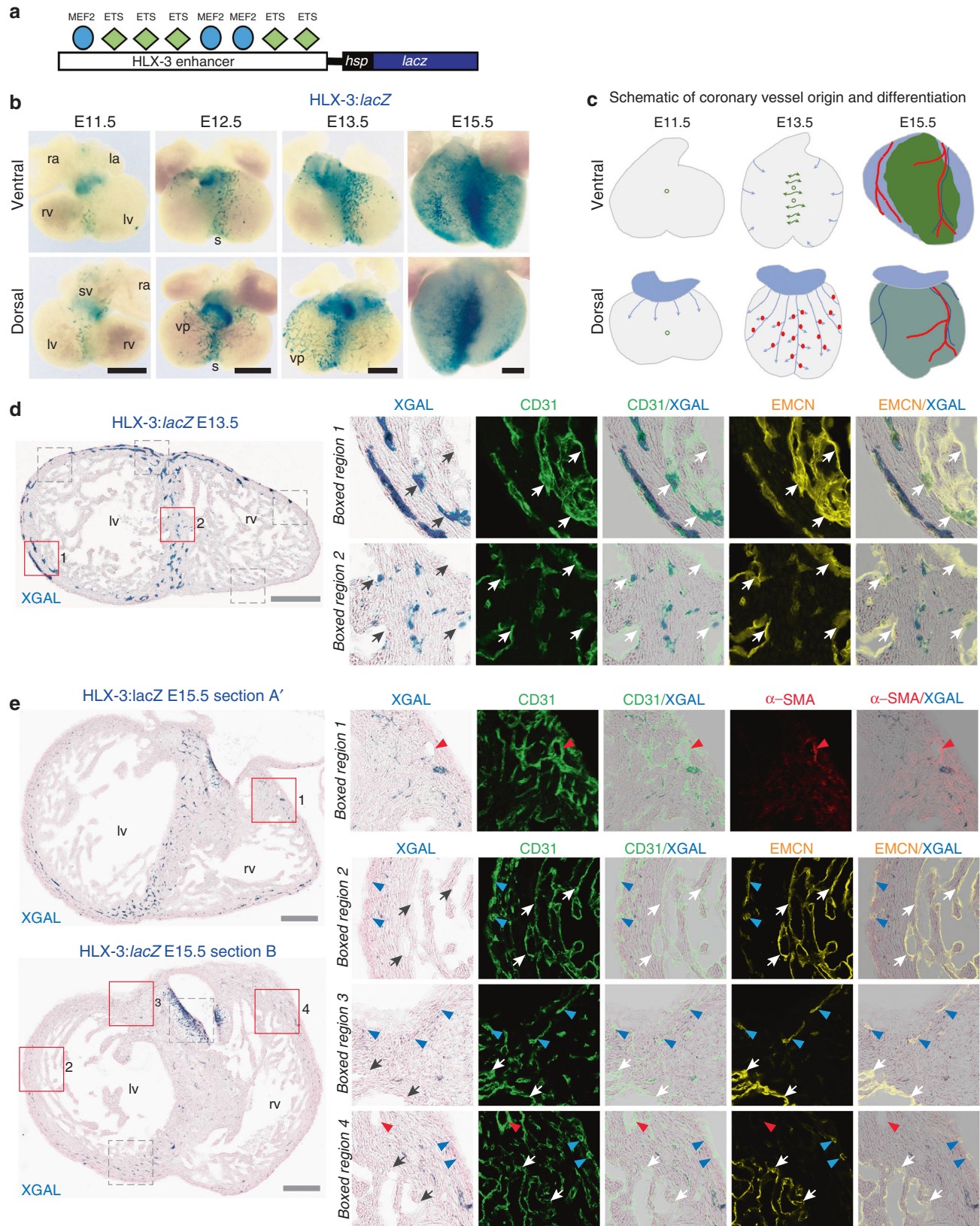

development (Fig. 1b–e and Supplementary Figs. 3 and 4). The activity of HLX-3 was similar in four different independent transgenic lines (Supplementary Fig. 5). In contrast with embryonic hearts transgenic for the ApjCreER;Rosa^mTmG lineage[10], which marks the SV-derived coronary vasculature, very little HLX-3-driven *lacZ* expression was seen at the dorsal base of the heart at E11.5, where coronary vessels first sprout from the SV (Fig. 1b, c). However, HLX-3:*lacZ* transgene activity was transiently detected in this vascular plexus at E12.5 and E13.5, and was detected in most ECs in the coronary vasculature at

**Fig. 1** The MEF2-dependent HLX-3 enhancer directs gene expression within the developing coronary vasculature. **a** Schematic of the HLX-3 enhancer:*lacZ* transgene. Coloured shapes represent verified MEF2 and ETS binding motifs within the human HLX-3 enhancer sequence, upstream of the *hsp68* silent minimal promoter and *lacZ* reporter gene[19]. **b** Whole-mount images of embryonic E11.5–E15.5 HLX-3:*lacZ* transgenic hearts show enhancer activity, as detected by blue X-gal staining, in the endothelium of selective coronary vessels. All images are from a single transgenic line and are representative of at least five biologically independent samples, little variation in staining pattern was seen. Supplementary Figure 5 shows comparative expression patterns in three other HLX-3:*lacZ* stable transgenic mouse lines. **c** Schematic summarising the current model of coronary vessel origin and differentiation, adapted from ref. [10,16,34]. Light blue represents SV-derived vascular plexus, green represents endocardium-derived intramyocardial vessels, turquoise colouring of the E15.5 dorsal side indicates presence of vessels from both origins. SV-derived arterial and pre-arterial vessels are indicated in red, SV-derived veins are indicated in dark blue. **d** Transverse section through a E13.5 HLX-3:*lacZ* transgenic heart. Immunostaining and X-gal staining was conducted and imaged sequentially on a single sample. HLX-3 enhancer activity (blue staining) is compared with pan-endothelial marker CD31 (green) and venous/endocardial marker EMCN (yellow). Grey boxed regions are shown in detail in Supplementary Fig. 3b. Representative of four biologically immunohistological independent experiments, with similar staining patterns seen in all. **e** Transverse section through a E15.5 HLX-3:*lacZ* transgenic heart. HLX-3 enhancer activity (blue) is compared with pan-endothelial marker CD31 (green), arterial marker α-SMA(red) and venous/endocardial marker EMCN (yellow). Two sections are shown, representing different levels in the heart from base to apex. Grey boxed regions are shown in Supplementary Fig. 4 alongside additional arterial and venous markers. Representative of seven biologically independent immunohistological experiments, with similar staining patterns seen in all. Black/white arrows indicate endocardium, red arrows indicate α-SMA-positive arteries, blue arrows indicate veins determined by EMCN expression and sub-epicardial location. ra, right atrium; la, left atrium; rv, right ventricle; lv, left ventricle; s, septum; vp, vascular plexus. Black scale bars represent 500 μm, grey scale bars represent 200 μm

E13.5. Only limited and punctate endocardial HLX-3 activity was seen by E13.5, although it was expressed more widely in endocardial cells at earlier time points (Fig. 1d and Supplementary Fig. 3). Expression in the SV-derived vasculature was not maintained through development, and little transgene expression was seen on the dorsal face by E15.5 (Fig. 1b–e and Supplementary Fig. 4). Whilst the precise time window of transgene activity cannot be defined due to the one-day half-life of β-gal activity[33], the reduction of HLX-3 activity correlates with the onset of arterial and venous differentiation in the SV-derived coronary plexus[34]. This suggests that this angiogenic pathway is transiently activated as SV-derived ECs lose their venous identity and expand down the dorsal side of the heart, and then switched off concurrent with further differentiation into mature arteries and veins.

In addition to transient activity in the SV-derived dorsal coronary plexus, HLX-3:*lacZ* activity was robustly detected in vessels within the septum from E11.5 (Fig. 1b–e). Unlike in the SV-derived vessels, this activity continued at later embryonic stages and expanded to vessels sprouting from the midline to the ventral face of the heart by E15.5. No further endocardial activity was detected, and HLX-3:*lacZ* expression rarely overlapped with markers for mature arteries or veins, although some HLX-3:*lacZ* expression was transiently detected in the valve cushions (Fig. 1b–e and Supplementary Fig. 4). The overall transgene expression pattern at E15.5 closely correlates with Nfatc1:Cre and the DACH1-positive, Apj:CreER-negative population, both of which represent endocardium-derived coronary vessels[10,12], and is consistent with the known role of VEGFA in endocardial-derived coronary vessels[12]. These results therefore strongly suggest that the VEGFA-MEF2 angiogenic regulatory pathway is active in endocardium-derived coronary endothelium throughout embryonic development, and indicate a role for MEF2 transcription factors as key regulators of coronary vessel differentiation.

It has been proposed that neonatal growth of new coronary vessels from the endocardium uses an alternative, non-angiogenic pathway distinct from that seen in the embryo[15]. However, we detected sustained HLX-3:*lacZ* expression in the neonatal and adult coronary vessels (Fig. 2 and Supplementary Fig. 6). Although we saw some variations in expression levels between adult age-matched hearts within single litters, the expression pattern and distribution of HLX-3:*lacZ* activity remained similar (Supplementary Fig. 6b). At neonatal stages (post-natal day (P)0), enhancer activity was detected throughout the vascularised portion of the myocardial wall although was more focused to the inner myocardial wall (Fig. 2b), and expression became limited to the inner myocardial wall by later stages (P8 and P56; Fig. 2c, d). Vessels within this region correspond to the putative second coronary vascular population originating from the endocardium[15], but it is unclear whether vessels within the outer myocardium wall utilise an alternative transcriptional pathway or are primarily quiescent at these stages. Together, these results suggest that classical VEGFA-stimulated angiogenesis is an important process in both pre- and post-natal coronary vessel formation, and in maintenance and homoeostasis of the coronary vessels during adult stages.

Although the HLX-3 enhancer requires MEF2 factor binding for activation, we cannot exclude the possibility that other transcriptional regulators also influence its expression in the heart. Consequently, we next investigated the cardiac activity of a second VEGFA-MEF2-dependent enhancer, Dll4in3. This enhancer, which regulates the Notch ligand *Dll4*, is directly bound and activated by MEF2 factors during sprouting angiogenesis, but shares little additional homology with HLX-3 beyond ETS motifs[7,18,19]. Unlike HLX-3, Dll4in3 is also independently activated in arterial ECs within the systemic vasculature by RBPJ (the transcriptional effector of Notch signalling) and SOXF transcription factors (Fig. 3a)[19]. As anticipated, mice transgenic for Dll4in3:*lacZ* expressed the reporter gene in regions associated with both SV-derived and endocardium-derived coronary vessels at early stages of coronary development (Fig. 3b). As with HLX-3:*lacZ*, some transgene expression was seen in the endocardium at E10.5, but was lost by E15.5 (Supplementary Fig. 7). However, the Dll4in3 enhancer had a wider field of activity compared to HLX-3 by E15.5, with distinct transgene expression seen in the endothelium of differentiating arteries in the outer myocardium on both the dorsal side and lateral regions of the ventral side of the heart (Fig. 3b and Supplementary Fig. 7b). Interestingly, no activity was seen in the valve cushions, suggesting that the HLX-3 valve cushion activity was either a secondary consequence of transgene insertion, or downstream of non-MEF2 factors activating HLX-3 but not Dll4in3.

Similar to HLX-3, activity of Dll4in3 was maintained after birth in vessels within the inner myocardial wall, but was also robustly active in neonatal arterial endothelium at P0 and P8 (Fig. 3b, c and Supplementary Fig. 8). Relatively little arterial activity was seen in adult hearts, instead transgene activity was principally limited to the inner myocardial wall as with HLX-3:*lacZ* (Supplementary Fig. 8c). The absence of arterial activity was

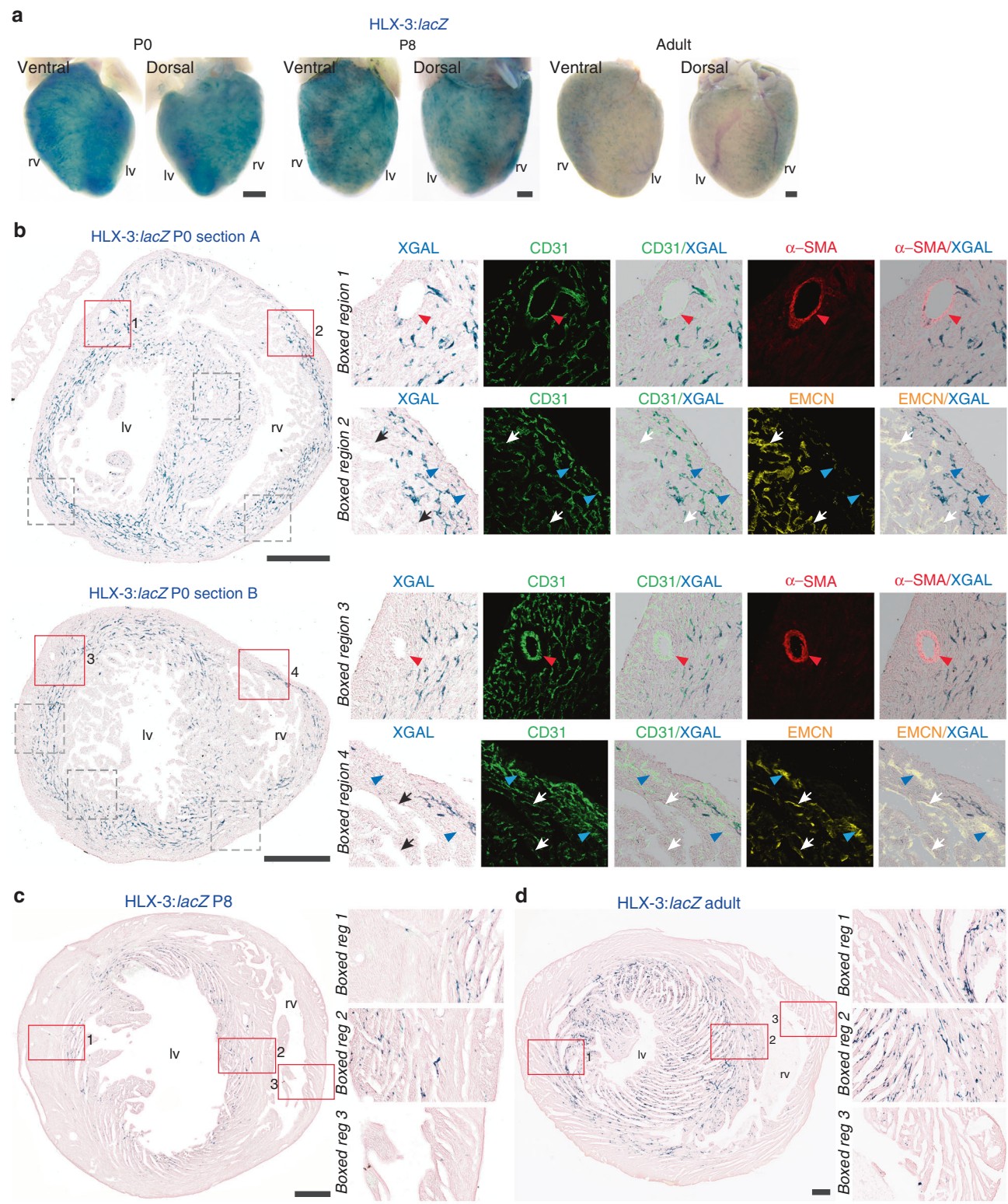

similar in other adult tissues, although both arterial and angiogenic activity of the Dll4in3:*lacZ* transgene could be reactivated in adult mice by adVEGFA (Supplementary Fig. 8d–e).

To test our hypothesised link between MEF2 binding and activity of the angiogenic sprouting pathway in early coronary formation, we next investigated the consequences of the loss of MEF2 binding in the Dll4in3 enhancer. In the systemic

vasculature, mutating the MEF2 sites in Dll4in3 (creating Dll4in3mutMEF2:*lacZ*, Fig. 3d) resulted in loss of angiogenic expression while maintaining transgene activity in arteries[19]. In the heart, expression of the Dll4in3mutMEF2:*lacZ* transgene was not seen at E11.5, and little activity was observed in the septum and ventral aspect of the coronaries throughout development (Fig. 3e and Supplementary Figs. 9 and 10). Punctate enhancer activity was, however, seen on the dorsal face at E13.5, correlating

**Fig. 2** HLX-3 enhancer activity is maintained in the coronary vasculature of the post-natal heart. **a** Whole-mount P0, P8 and adult HLX-3:*lacZ* transgenic hearts demonstrate enhancer activity in the post-natal heart as detected by blue X-gal staining. All images are from a single transgenic line and are representative of at least five biologically independent samples, little variation in staining pattern was seen apart from in adult heart, an example of the weaker staining intensity is shown in Supplementary Fig. 6b. **b** Transverse sections sequentially from base to apex through a P0 HLX-3:*lacZ* heart. HLX-3 enhancer activity is detected by blue staining in coronary vessels located in the intramyocardium and inter-ventricular septum. Enhancer activity (blue) is compared to pan-endothelial marker CD31 (green), the arterial marker α-SMA (red) and the venous/endocardial marker EMCN (yellow). Grey boxed regions are shown in Supplementary Fig. 6a. Representative of three biologically independent immunohistological experiments, with similar staining patterns seen in all. **c, d** Transverse sections through a P8 (**c**) and adult (**d**) HLX-3:*lacZ* transgenic hearts. Continued HLX-3:*lacZ* activity (blue) is seen in small vessels of the inner myocardium and inter-ventricular septum. Representative of over five biologically independent samples, examples of lower intensity staining seen in less than half adult hearts is shown in Supplementary Fig. 6b. Black/white arrows indicate the endocardium, red arrowheads indicate arteries and blue arrowheads indicate veins. rv, right ventricle; lv, left ventricle. Black scale bars represent 500 μm

with a SV-derived pre-arterial coronary EC population described by Su et al.[34] (Fig. 3e). At later embryonic and early post-natal stages, Dll4in3mutMEF2 activity was again missing in the microvasculature but was seen in most mature arterial coronary vessels in the outer myocardium (Fig. 3e, f and Supplementary Figs. 9 and 10a). As with Dll4in3, arterial activity was lost in the adult mouse (Supplementary Fig. 10b). These results further demonstrate a clear role for the VEGFA-MEF2 pathway in gene activity in endocardial-derived coronary vasculature.

**SOXF/RBPJ pathway is active in SV-derived coronary arteries.** Marker and single-cell analysis has suggested that coronary arteries develop from a subset of SV-derived coronary vessels which lose the expression of venous markers and progressively acquire arterial ones[11,34]. The activity of Dll4in3 and Dll4in3-mutMEF enhancers in the mature coronary arteries indicates that the pathways regulating these enhancers may play an important role in arterial acquisition in the SV-derived coronary vasculature. To confirm this, we studied the cardiac activity of a second *Dll4* enhancer, Dll4-12. Like Dll4in3, the Dll4-12 enhancer is directly bound by ETS, RBPJ and SOXF factors (Fig. 4a) and is expressed in arterial ECs[7]. However, it does not bind MEF2 and is not active during sprouting angiogenesis[7,19]. The expression of the Dll4-12:*lacZ* transgene in the coronary vasculature in two independent transgenic lines was strikingly similar to that of Dll4in3mutMEF2 (Fig. 4 and Supplementary Fig. 11). Unlike Dll4in3 and HLX-3, enhancer activity was not seen in the endocardium at E10 (Supplementary Fig. 11b). In the heart, Dll4-12:*lacZ* expression was first detected at around E13.5, and was initially focused on the dorsal aspect towards the base of the heart (Fig. 4b). This region corresponds to the SV-derived vascular plexus. The punctate and discontinuous expression pattern of the Dll4-12:*lacZ* transgene closely resembles that described for SV-derived pre-arterial cells[34], suggesting that the SOXF/RBPJ arterial transcriptional programme is activated as this cell population switches fate from venous to arterial. As with Dll4in3mutMEF2, Dll4-12:*lacZ* activity at E15.5 was specific to arterial endothelium in the outer myocardium of both sides of the heart, where it was maintained after birth although lost in adult tissues (Fig. 4c, d and Supplementary Figs. 11 and 12). Expression in the septum was restricted to differentiated arteries (Fig. 4d and Supplementary Figs. 11c and 12a). These results indicate that RBPJ and SOXF factors, which bind both Dll4in3 and Dll4-12 enhancers, can direct independent gene activity in developing coronary arteries from the earliest stages of arterial differentiation within the SV-derived vascular plexus.

To determine the relative contributions of RBPJ and SOXF to this pattern, we next examined an enhancer driven by SOXF and ETS, but not RBPJ. SOXF factors activate expression of the Notch receptor *Notch1* through the NOTCH1+16 enhancer without requiring RBPJ binding[21] (Fig. 5a). Since Notch expression is itself required for RBPJ activity, this positions SOXF factors

upstream of Notch/RBPJ in arterial differentiation[21]. The expression of the NOTCH1+16:*lacZ* transgene in the early heart was more extensive than that seen for the previously examined enhancers, and correlated closely with SV-derived coronary vessels[10] (Fig. 1c, Fig. 5b–d and Supplementary Fig. 13). Unlike Dll4mutMEF and Dll4-12 enhancers, this activity was not restricted to the pre-arterial population, reflecting the similarly non-specific expression of endogenous *Notch1* in the coronaries at this time point[34,35]. Unlike HLX-3, little expression was detected in the septum or ventral side, suggesting activity was limited to SV-derived coronary endothelium (Fig. 5b). Similar to its expression in the systemic vasculature[21], NOTCH1+16 activity became progressively restricted to the arterial coronary vessels as coronary blood flow was fully established (Fig. 5c and Supplementary Fig. 13a, b). Post-natal expression was seen in both mature arterial and lymphatic vessels, the latter consistent with the role of Notch1 and SoxF factors in lymphatic differentiation[36,37] (Fig. 5d and Supplementary Fig. 13c–e). These data suggest that a SOXF-driven arterial programme is initially active in all SV-derived coronary vessels, and that the pre-flow signal for ECs to differentiate into pre-arterial cells requires RBPJ.

**Most coronary ECs activate a single regulatory pathway.** The results presented so far suggest that the VEGFA-MEF2 angiogenic and SOXF/RBPJ regulatory pathways are active in distinct populations of ECs within the heart. To directly test this hypothesis, we crossed the Dll4-12:*lacZ* transgenic mice with a line expressing a HLX-3:tdTomato transgene, in which the *lacZ* reporter gene is replaced with tdTomato (Fig. 6a). HLX-3:tdTomato transgenic mice had similar spatial distribution of reporter gene activity in coronary vessels to HLX-3:*lacZ*, although tdTomato expression was more intense in positive vessels, and all lines demonstrated an ectopic activity in some regions of myocardium never seen in the HLX-3:*lacZ* mice (Supplementary Fig. 14).

At E15.5, a time point when both pathways were robustly active within the heart, cross-sectional analysis of most coronary vessels showed either activity of a single transgene or no transgene activity (Fig. 6 and Supplementary Fig. 14). However, we also found that a subset of coronary vessels, predominantly in arterial positions, expressed both transgenes (Fig. 6b, c). In approximately half of these cases, there was no overlap in transgene activity at the cellular level, with distinct ECs within a single arterial cross-section expressing either the Dll4-12 or HLX-3 transgenes in a mosaic manner. In other such vessels, we found some ECs expressing both transgenes (Fig. 6b, c and Supplementary Fig. 14). These observations concur with the separate analyses of HLX-3:*lacZ* and Dll4-12:*lacZ* lines in the coronaries, where we saw occasional arterial ECs expressing HLX-3:*lacZ* or lacking Dll4in3mutMEF2:*lacZ* and Dll4-12:*lacZ* (Supplementary Figs. 4, 10–12). Due to the half-life of the reporter genes, we were unable to establish whether these represent a single EC in which

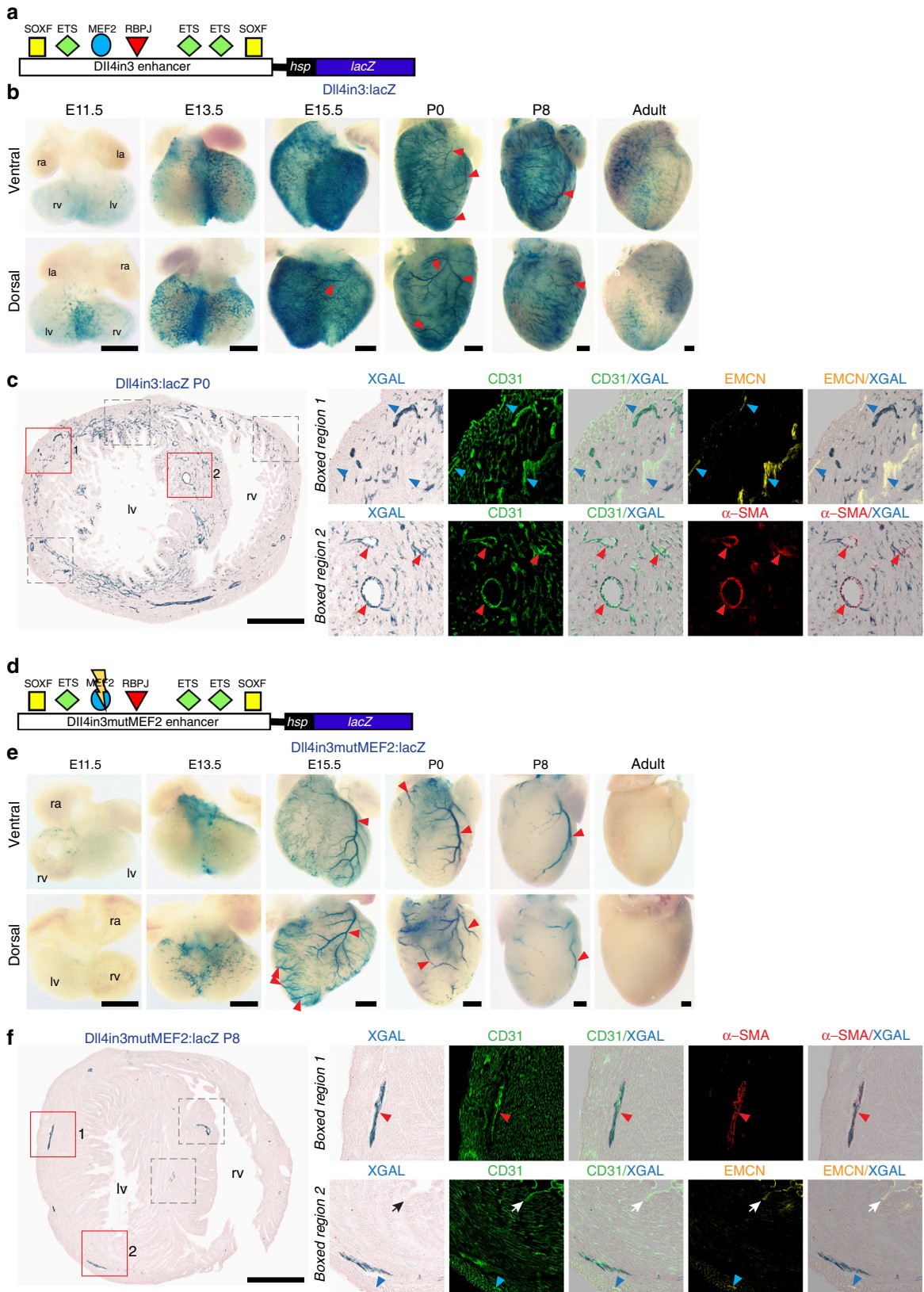

both the SOXF/RBPJ and VEGFA-MEF2 pathways are active, or the transitioning of ECs from angiogenic to arterial identity, a phenomenon reported in the systemic vasculature[22,38,39]. These results also align with observations from Cre-driven lineage tracing: both the SV-associated ApjCreER and endocardial-associated Nfatc1Cre labelled coronary vessels on the dorsal side

of E15.5 hearts, with the cumulative percentage of cells labelled by either pathway significantly above 100%[10].

**BMP/SMAD pathway is active in SV-derived coronary veins**. The initial sprouts of the coronary vessel plexus originate from

**Fig. 3** MEF2 factor binding is required for activity of the Dll4in3 enhancer in angiogenic coronary vessels. **a** Schematic of the Dll4in3 enhancer:*lacZ* transgene, with verified binding motifs for SOXF, ETS, MEF2 and RBPJ transcription factors represented by coloured shapes[7,19]. **b** Whole-mount images of Dll4in3:*lacZ* transgenic hearts show enhancer activity, as detected by blue X-gal staining, in the endothelium of selective coronary vessels. Transgene expression was detected in coronary arteries from E15.5 (red arrowheads). All images are from a single transgenic line and are representative of at least five biologically independent samples, little variation in staining pattern was seen. **c** Transverse section through a P0 Dll4in3:*lacZ* heart. Enhancer activity (blue) is compared to pan-endothelial marker CD31 (green), the arterial marker α-SMA (red) and the venous/endocardial marker EMCN (yellow). Representative of three biologically independent immunohistological experiments, with similar staining patterns seen in all. Grey boxed regions are shown in Supplementary Fig. 8. Sections of E15.5 hearts are shown in Supplementary Fig. 7b. **d** Schematic of the mutated Dll4in3mutMEF2 enhancer:*lacZ* transgene, which is unable to bind MEF2 factors due to point mutation (represented by lightning strike)[19]. **e** Whole-mount images of Dll4in3mutMEF2:*lacZ* transgenic hearts show an absence of enhancer activity (blue staining) on the ventral aspect and inter-ventricular septum, activity is seen in punctate regions of the dorsal vascular plexus at E13.5 and in coronary arteries (red arrowheads) at E15.5, P0 and P8. All images are from a single transgenic line and are representative of at least five biologically independent samples, little variation in staining pattern was seen. **f** Transverse section through a P8 Dll4in3mutMEF2:*lacZ* heart. Enhancer activity is compared to pan-endothelial marker CD31 (green), the arterial marker α-SMA (red) and the venous/endocardial marker EMCN (yellow). Representative of four biologically independent immunohistological experiments, with similar staining patterns seen in all. Grey boxed regions are shown in Supplementary Fig. 10a. Sections of E15.5 hearts are shown in Supplementary Fig. 9. Black/white arrows indicate the endocardium, red arrowheads indicate arteries and blue arrowheads indicate veins. ra, right atrium; la, left atrium; rv, right ventricle; lv, left ventricle. Black scale bars represent 500 μm

venous cells within the SV[11]. Previous studies in the systemic zebrafish and mouse vasculature have clearly implicated BMP signalling in venous identity and sprouting[22,40–42]. Although tamoxifen-induced depletion of SMAD4 beginning at E10.5 did not affect the early migration of the SV-plexus over the heart[43], the SMAD4 depletion occurred too late to affect the first sprouts from the SV. Consequently, we next investigated the role of BMP-driven venous regulatory pathways in coronary vessel formation. Our laboratory has recently characterised an enhancer for the venous gene *Ephb4*. This enhancer, termed Ephb4-2, is specifically active in venous endothelium throughout development downstream of ALK3-BMP signalling via direct binding of the SMAD1/5:SMAD4 transcriptional activators[22] (Fig. 7a). Consistent with the venous origin of SV-derived coronary sprouts, we saw robust activity of the Ephb4-2:*lacZ* transgene at the dorsal base of the heart at E11.5, the location where coronary vessels first emerge from the SV (Fig. 7b)[10]. However, transgene activity on this dorsal face was significantly diminished by E13.5, and from E15.5 activity was specifically detected only in a limited number of superficial coronary vessels primarily found on the dorsal aspect (Fig. 7b). This correlated with the venous markers EPHB4 and EMCN in superficial vessels, but did not overlap with EMCN/EPHB4-positive endocardium (Fig. 7c, d and Supplementary Fig. 15). No activity was seen in arterial cells, although limited transgene expression was seen in restricted regions of the outflow tract (Fig. 7c). These results correlate with previous observations into the differentiation of SV-derived coronary vessels, which found that their venous identity is lost by E12.5, before reactivating in superficial vessels as they form the mature coronary veins[11]. As with the arterial enhancers, little Ephb4-2:*lacZ* activity was seen in adult tissues (Supplementary Fig. 15). Similar patterns of transgene activity were also observed in a second BMP-SMAD1/5:SMAD4 driven venous enhancer transgene, CoupTFII-965:*lacZ*, although this transgene (and endogenous *Coup-TFII/Nr2f2*) was also expressed in both systemic and coronary lymphatic vessels[22] (Supplementary Fig. 16). These results, therefore, indicate a role for BMP signalling in the formation of the coronary vasculature, and suggest that a similar ALK3-BMP-SMAD1/5 pathway may regulate both systemic and coronary venous differentiation.

**ECs in injured neonatal heart activate developmental pathways.** Previous investigations into the origins of neovascular growth after ischaemic injury in mice implicated both angiogenesis and de novo vessel formation, but did not establish the regulatory pathways behind these processes[44,45]. Although it is often presumed that ischaemic neovascularisation in the damaged heart

uses similar pathways to those governing developmental coronary growth, this has not yet been studied directly. We therefore investigated whether the VEGFA-MEF2 driven angiogenic, SOXF/RBPJ driven arterial and BMP-SMAD1/5 driven venous pathways were involved in coronary vessel growth in response to injury. First, we investigated the response to ischaemic injury at the early post-birth time-period, during which all three pathways were active in uninjured hearts (Figs. 2, 4 and 7). To simulate MI, we performed permanent ligation of the left anterior descending coronary artery (LAD) in the hearts of neonatal transgenic mice at P1, a stage at which they can fully regenerate[46]. Two days after MI, activity of all three regulatory pathways was detected in ECs within the infarct and border zone (Fig. 8 and Supplementary Fig. 17), indicating that the developmental pathways controlling the initial formation of the coronary vessels are active in the injured neonate. We next investigated whether this activation was restricted to the highly regenerative neonatal stage, repeating the MI in P7 hearts[46] (Fig. 9). Although we hypothesised we would see different patterns of activity as P7 hearts cannot fully regenerate or form arterial-derived collateral vessels after MI[47], all three developmental coronary regulatory pathways were still robustly active around the infarct (Fig. 9). This indicates that the activation of developmental vascular pathway activity was not restricted to hearts with the ability to regenerate myocardium. Normal expression levels of the transgenes were not detected in remote regions of the hearts after MI as they were in control hearts at both time points, suggesting that ischaemic injury may have activated regulatory pathways in one region of the heart at the expense of other regions.

**MEF2 pathway is not activated in the injured adult heart.** We next investigated whether these coronary vascular developmental pathways could be reactivated in adult hearts after MI. Using a similar LAD ligation model, we investigated the behaviour of the angiogenic HLX-3, arterial Dll4-12 and venous Ephb4-2 enhancers 2, 7 and 14 days post-MI. In contrast to the neonatal hearts, we detected no increase in HLX-3 activity after MI at any time point (Fig. 10a, Supplementary Figs. 18 and 19). In fact, HLX-3 activity was detected elsewhere in the injured hearts but was specifically absent in the infarct and border zone in all hearts examined at all time points, including in the inner myocardial wall. Although HLX-3:*lacZ* transgene expression intensity showed some variation in adult hearts (Supplementary Figs. 6b and 18), the complete lack of activation around the MI was consistently observed in all experiments ($n = 12$ biologically independent experiments). This expression pattern contrasts with the behaviour of angiogenic VEGFA-MEF2 driven enhancers in other

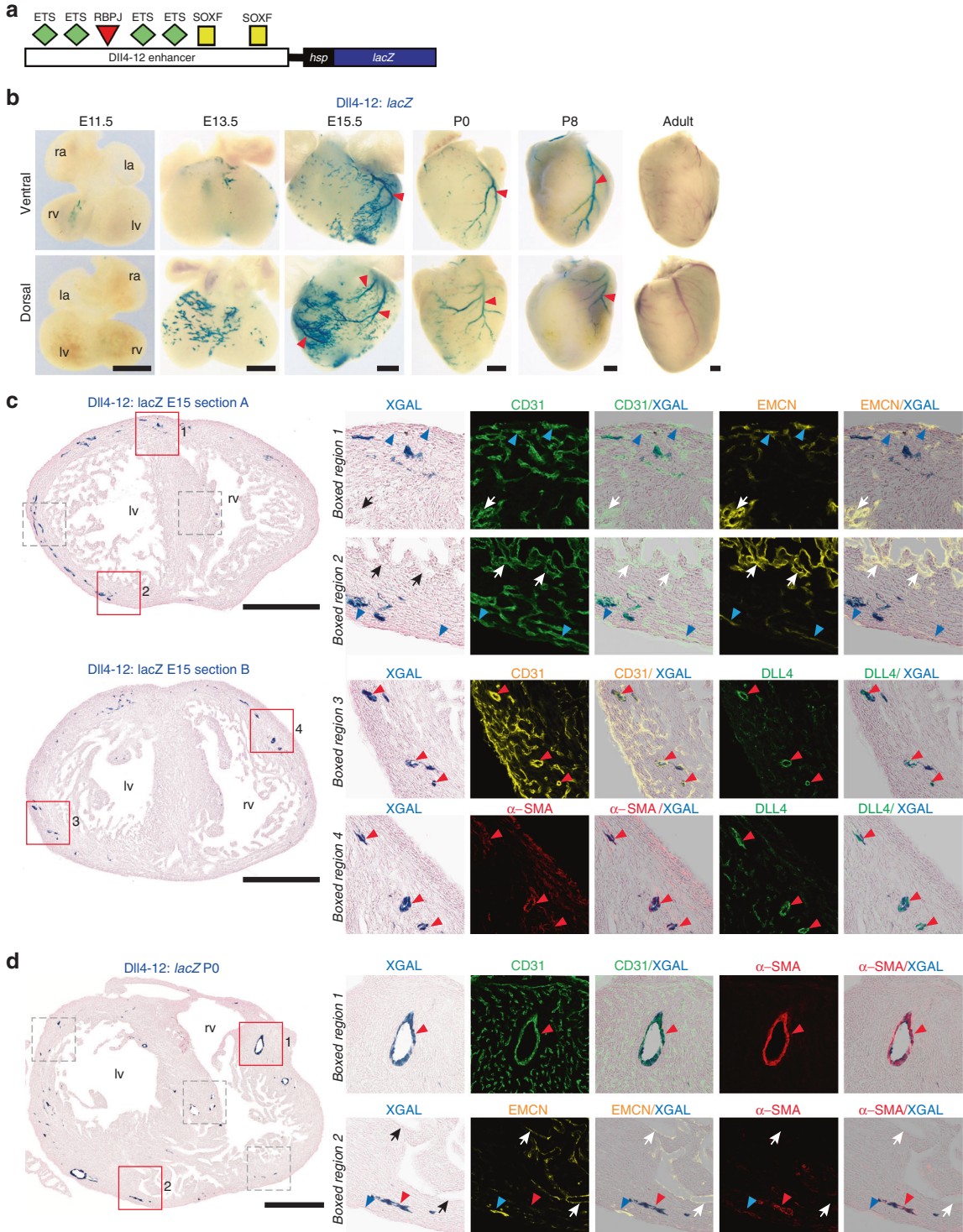

**Fig. 4** The SOXF/RBPJ-dependent Dll4-12 enhancer is specifically active in SV-derived coronary arteries. **a** Schematic of the Dll4-12 enhancer:*lacZ* transgene, with verified binding motifs for ETS, RBPJ and SOXF transcription factors represented by coloured shapes[19]. **b** Whole-mount images and transverse sections of Dll4-12:*lacZ* transgenic hearts demonstrate selective enhancer activity (blue staining) in coronary arterial endothelium. All images are from a single transgenic line and are representative of at least five biologically independent samples, little variation in staining pattern was seen. Supplemental Figure 11 shows comparative expression patterns in a second Dll4-12:*lacZ* stable transgenic mouse line. **c**, **d** Transverse sections through Dll4-12:*lacZ* E15.5 (**c**) and P0 (**d**) hearts show Dll4-12 enhancer activity (blue) compared to pan-endothelial marker CD31 (green in top panel, yellow in lower panel), the arterial markers α-SMA (red) and DLL4 (green) and the venous/endocardial marker EMCN (yellow). Representative of eight (**c**) and five (**d**) biologically independent immunohistological experiments, with similar staining patterns seen in all. Grey boxed regions are shown in Supplementary Figs. 11c and 12a. Black/white arrows indicate the endocardium, red arrowheads indicate arteries and blue arrowheads indicate veins. ra, right atrium; la, left atrium; rv, right ventricle; lv, left ventricle. Black scale bars represent 500 μm

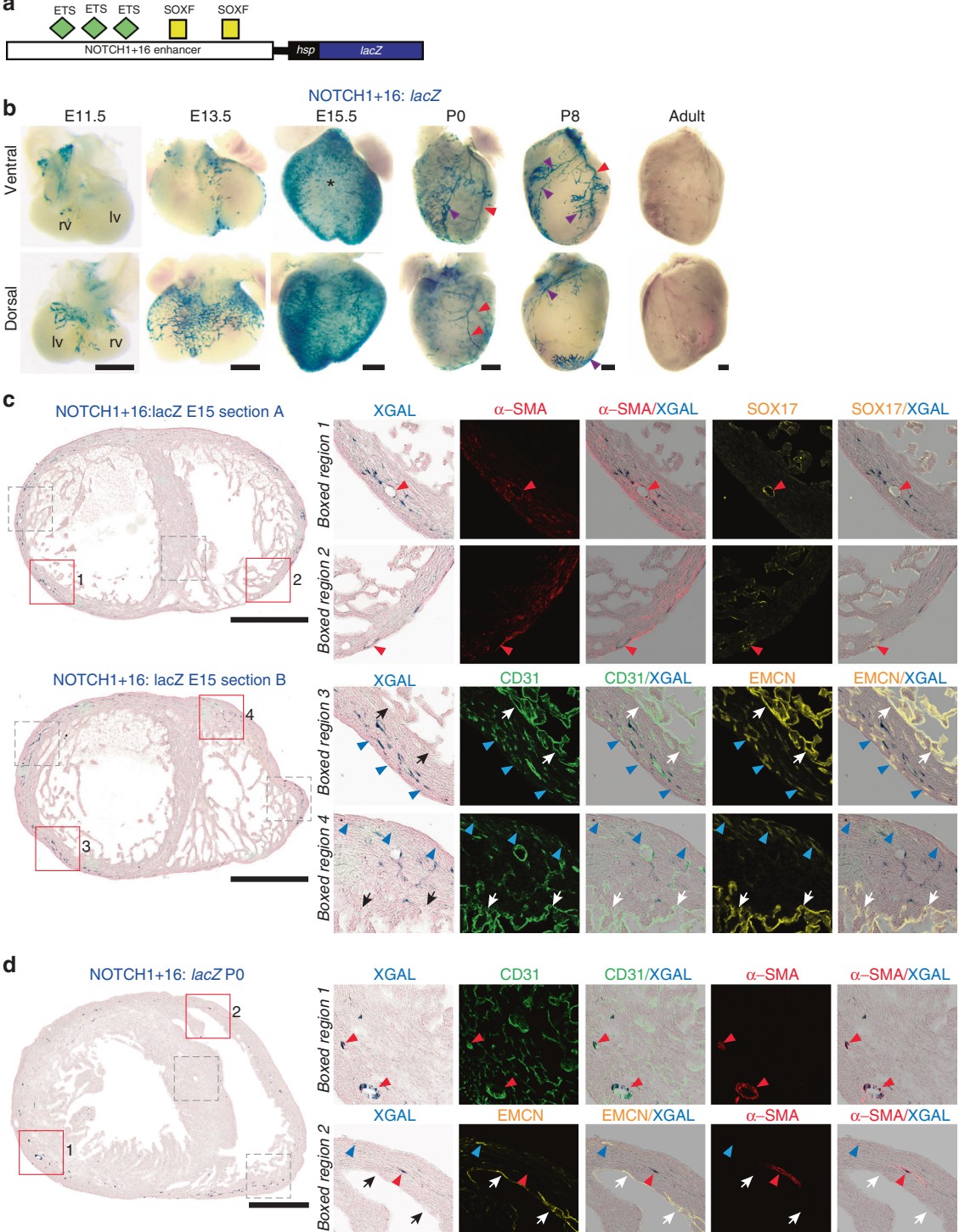

**Fig. 5** The SOXF-dependent NOTCH1+16 enhancer is restricted to SV-derived vessels. **a** Schematic showing the NOTCH1+16 enhancer:*lacZ* transgene, with verified binding motifs for ETS and SOXF transcription factor represented by coloured shapes[21]. **b** Whole-mount images of NOTCH1+16:*lacZ* transgenic hearts. NOTCH1+16 enhancer activity (blue staining) is not seen on the ventral aspect of the developing heart (*), is seen in the vascular plexus on the dorsal aspect in the embryonic heart, and is restricted to coronary arteries and lymphatic vessels (red and purple arrowheads) at P0 and P8. All images are from a single transgenic line and are representative of at least five biologically independent samples, little variation in staining pattern was seen. **c, d** Transverse sections through NOTCH1+16:*lacZ* E15.5 (**c**) and P0 (**d**) hearts show NOTCH1+16 enhancer activity (blue) compared to pan-endothelial marker CD31 (green), the arterial markers α-SMA (red) and SOX17 (green) and the venous/endocardial marker EMCN (yellow). Representative of five (**c**) and four (**d**) biologically independent immunohistological experiments, with similar staining patterns seen in all. Grey boxed regions are shown in Supplementary Fig. 13b, c. Black/white arrows indicate the endocardium, red arrowheads indicate arteries, blue arrowheads indicate veins, purple arrowheads indicate lymphatics. rv, right ventricle; lv, left ventricle. Black scale bars represent 500 μm

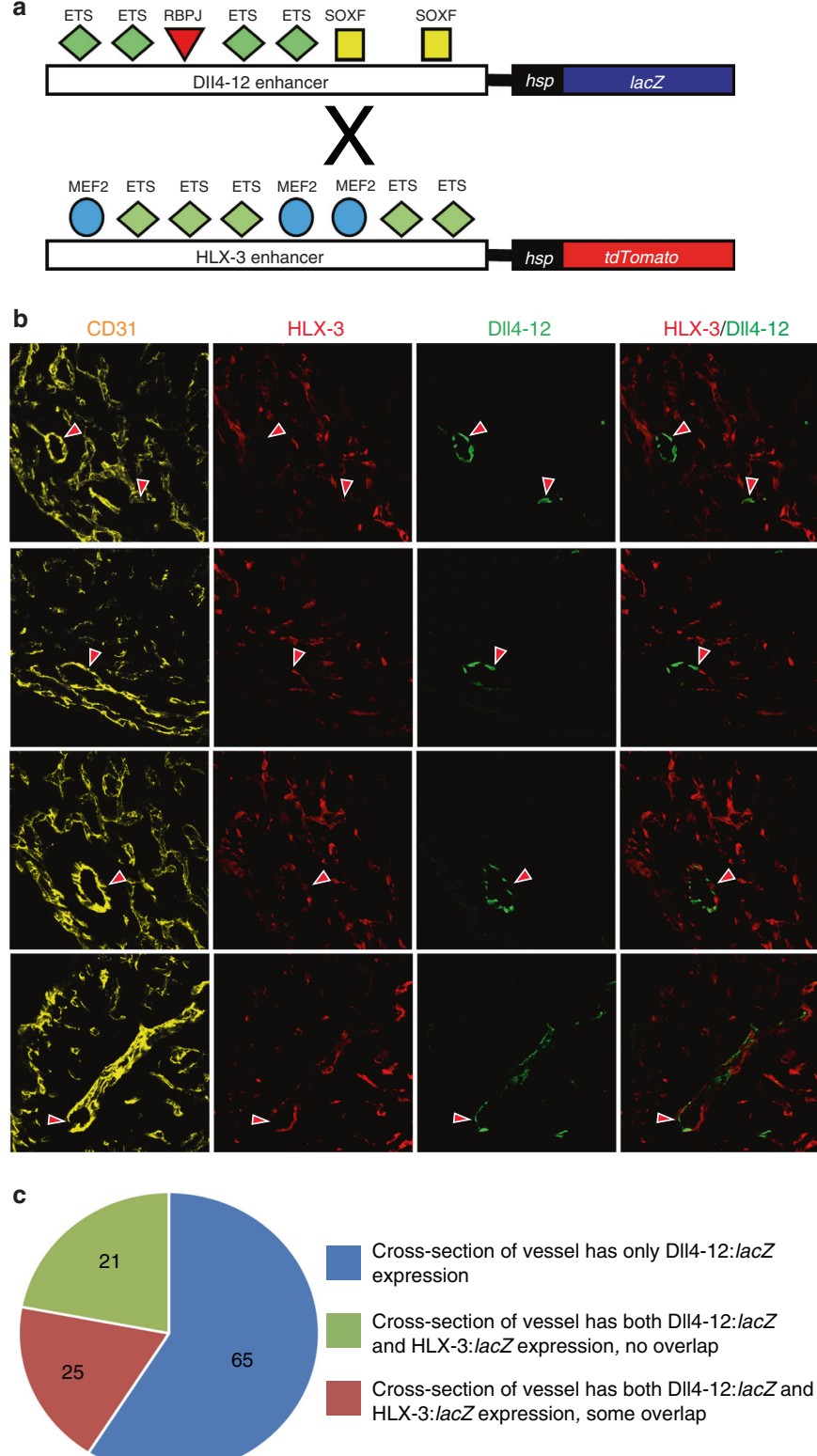

**Fig. 6** Direct comparison of HLX-3 and Dll4-12 enhancer activities in coronary vessels at E15.5. **a** Schematic showing the Dll4-12:*lacZ* and HLX-3:*tdTomato* transgenes, with coloured shapes representing verified transcription factor binding sites. E15.5 hearts were collected from mice transgenic for both transgenes. **b** Immunostaining showing transgene activity comparative to CD31 staining in the ventricular wall of E15.5 Dll4-12:*lacZ*;HLX-3:*tdTomato* transgenic hearts. Images showing whole heart images from which these are extracted can be seen in Supplementary Fig. 14a. Arterial vessels primarily expressed the Dll4-12:lacZ transgene (green, as detected by anti-ß-gal antibody) but some arterial-located ECs also showed HLX-3:tdTomato activity (red, as detected by anti-tdTomato antibody). **c** Quantification of enhancer activity in arterial position Dll4-12:*lacZ*-positive vessels. Numbers represent number of vessels investigated from four biologically independent E15.5 hearts at multiple levels of heart

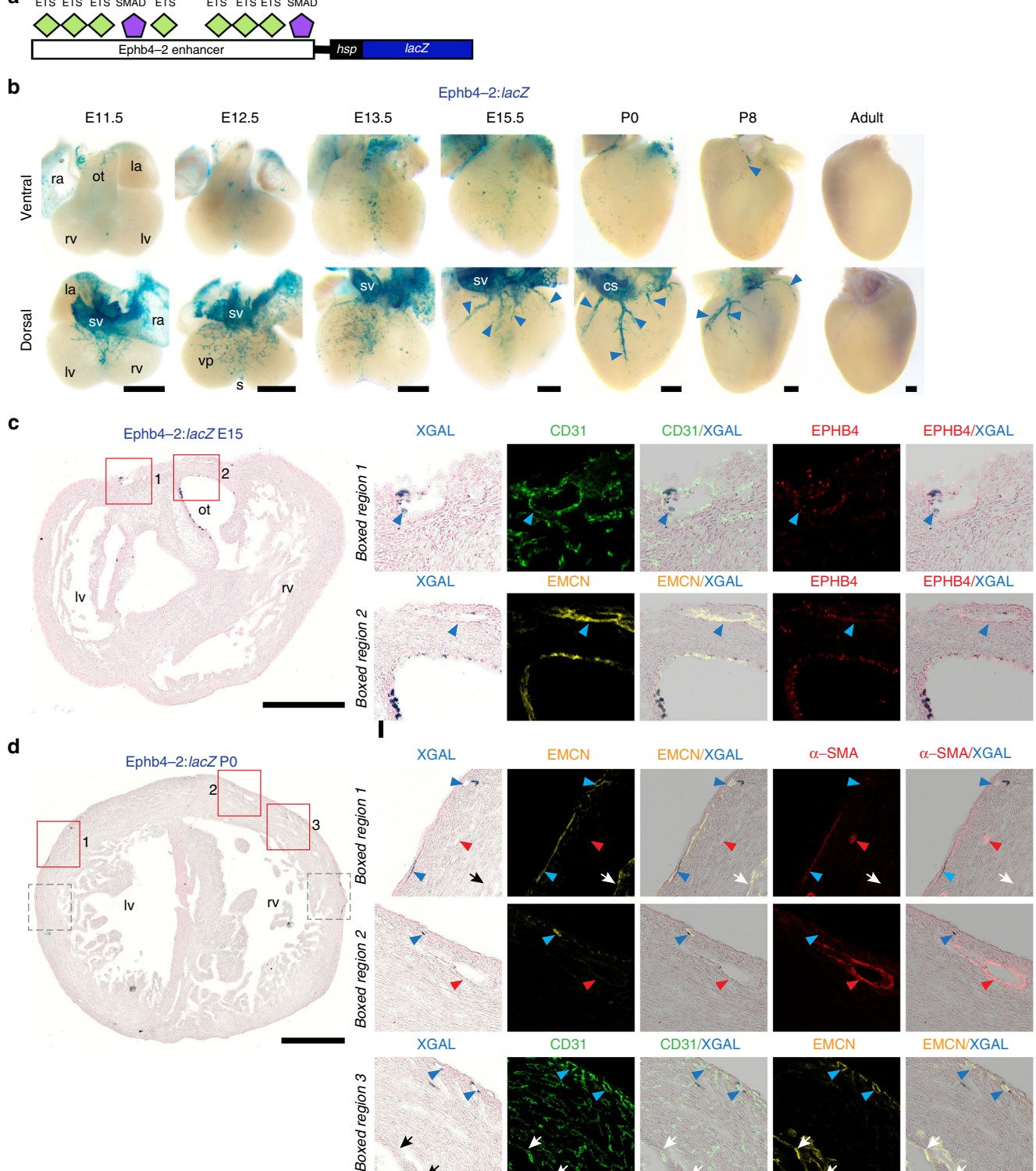

**Fig. 7** The vein-specific EphB4-2 enhancer is active in the SV, SV-derived vascular plexus and in mature coronary veins. **a** Schematic showing the EphB4-2 enhancer:*lacZ* transgene, with verified binding motifs for ETS and SMAD1/5 transcription factors represented by coloured shapes[22]. **b** Whole-mount images of embryonic and post-natal EphB4-2:*lacZ* transgenic hearts, showing strong enhancer activity in the SV and the early SV-derived plexus on the dorsal aspect. From E15.5, activity is restricted to the coronary veins (blue arrowhead). All images are from a single transgenic line and are representative of at least five biologically independent samples, little variation in staining pattern was seen. **c**, **d** Transverse sections through E15.5 (**c**) and P0 (**d**) EphB4-2:*lacZ* heart. Enhancer activity (blue) is compared to pan-endothelial marker CD31 (green), venous marker EPHB4 (red in top panel), venous/endocardial marker EMCN (yellow) and arterial markers α–SMA (red in middle panel). Representative of six (**c**) and three (**d**) biologically independent immunohistological experiments, with similar staining patterns seen in all. Grey boxed regions shown in Supplementary Fig. 15a. Black/white arrows indicate the endocardium, red arrowheads indicate arteries, blue arrowheads indicate veins. ra, right atrium; la, left atrium; rv, right ventricle; lv, left ventricle; sv, sinus venosus; ot, outflow tract; vp, vascular plexus; s, septum; cs, coronary sinus. Black scale bars represent 500 μm

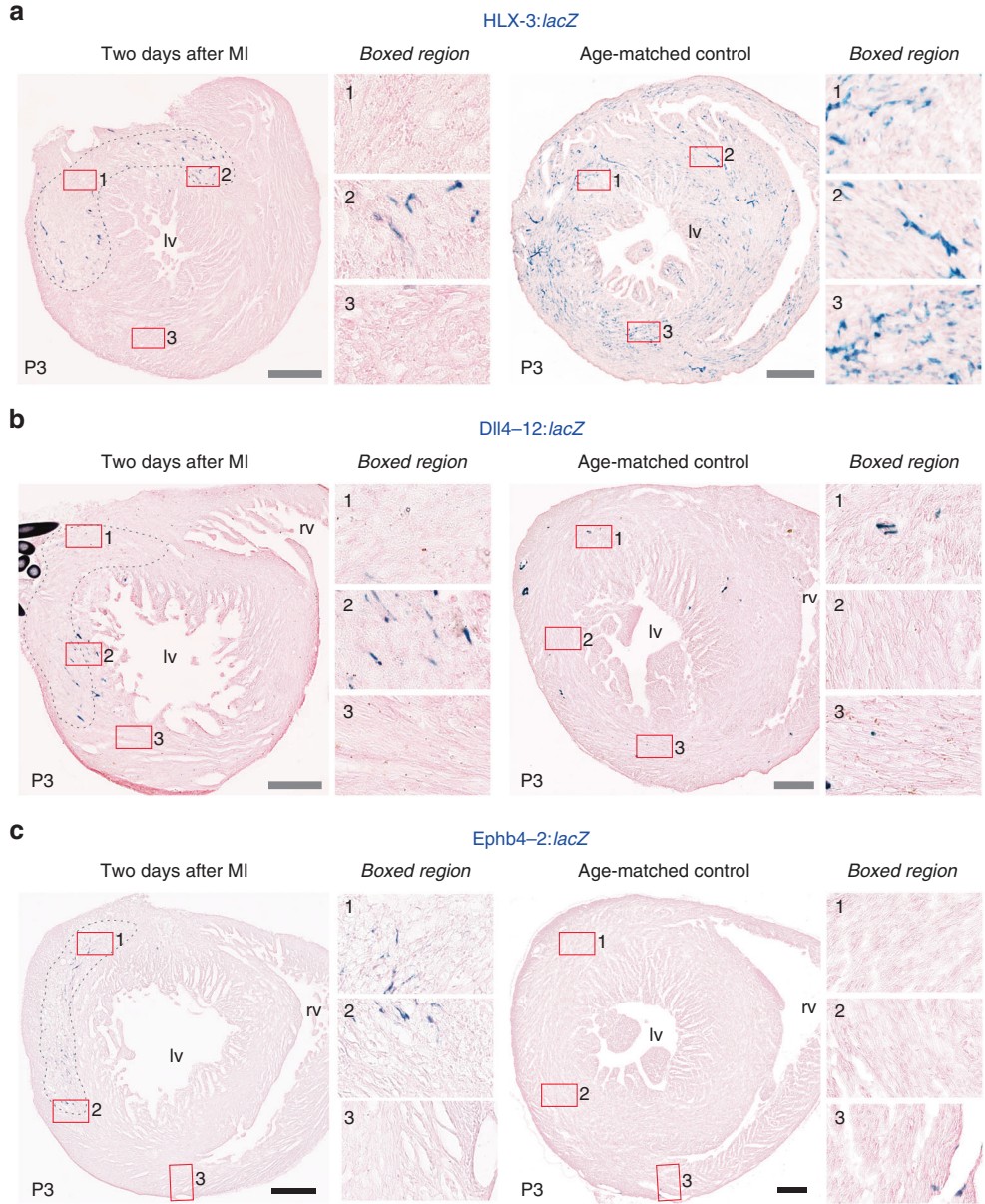

**Fig. 8** Developmental vascular regulatory pathways are reactivated during neovascularisation in the regenerative P1 neonatal heart after MI. **a–c** Permanent ligation of the left anterior descending coronary artery was performed on transgenic neonatal hearts to mimic MI. Representative sections through P3 HLX-3:*lacZ* (**a**, n = 6 biologically independent animals), Dll4-12:*lacZ* (**b**, n = 5 biologically independent animals) and Ephb4-2:*lacZ* (**c**, n = 4 biologically independent animals) transgenic hearts collected two days after MI surgery show enhancer activity in ECs surrounding the injured tissue. Dashed grey lines indicate the injured myocardium, and boxed regions show magnified views within the injury (box 1), at the injury border zone (box 2) and at a remote region (box 3). Sections through age-matched control hearts are shown alongside, with comparable boxed regions selected for the magnified views. lv, left ventricle; rv, right ventricle. Grey scale bars represent 200 μm

adult tissues: both HLX-3:*lacZ* and Dll4in3:*lacZ* activity was reactivated during tumour neovascularization, neovascular growth into a matrigel plug, and neoangiogenesis in response to administration of adVEGFA into the mouse ear[19] (Supplementary Figs. 8e and 20). This indicates that the VEGFA-MEF2 angiogenic pathway is specifically inactive during neovascularisation post-MI in adult hearts comparative to other adult settings, and suggests that intrinsic angiogenesis may be repressed in these ischaemic regions. The absence of HLX-3:*lacZ* expression did not correlate with loss of MEF2 factor expression, as MEF2C was detected in the vasculature surrounding infarcts after MI in both neonatal and adult hearts (Supplementary Fig. 21). Since the VEGFA-MEF2 angiogenic pathway is strongly influenced by

epigenetic modification of MEF2 factors[19], these observations suggest that repression of the ability of MEF2 factors to activate transcription may play a role in the loss of angiogenic HLX-3 activity in the adult ischaemic heart.

The expression of the arterial Dll4-12 and venous Ephb4-2 enhancers did not show the same patterns of repression. Although no enhancer activity was detected in the sham controls and in regions away from the infarct, both the Dll4-12:*lacZ* and Ephb4-2:*lacZ* transgenes were expressed in a small number of cells around the infarct 2 days after injury, and in many more cells after 7 days (Fig. 10 and Supplementary Fig. 19). Some expression remained two weeks post-MI (Supplementary Fig. 19). However, few of the enhancer-

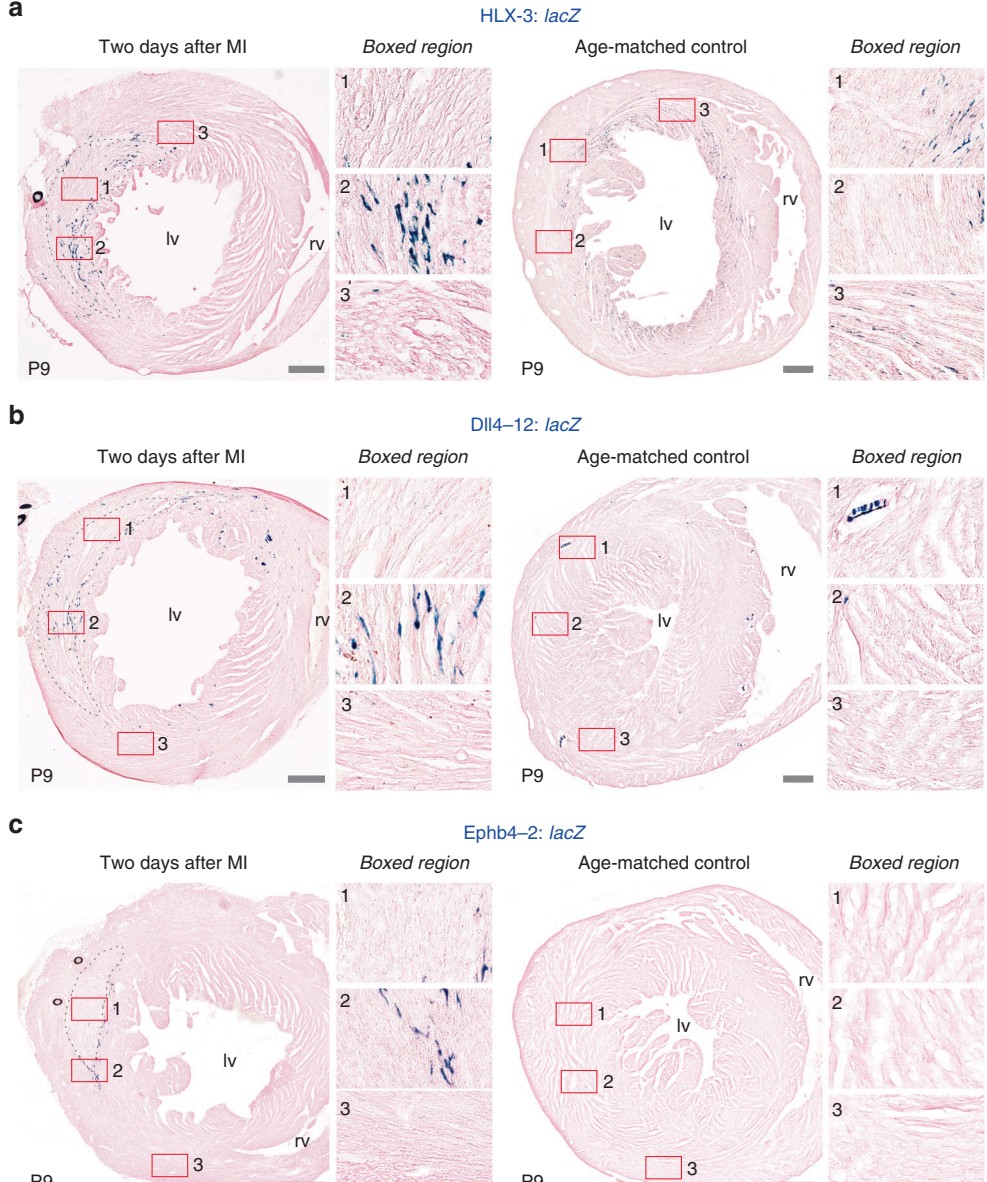

**Fig. 9** Developmental vascular regulatory pathways are reactivated during neovascularisation in the less regenerative P7 neonatal heart after MI. **a–c** Permanent ligation of the left anterior descending coronary artery was performed on transgenic P7 hearts to mimic MI. Representative sections through P9 HLX-3:*lacZ* (**a**, n = 6 biologically independent animals), Dll4-12:*lacZ* (**b**, n = 5 biologically independent animals) and Ephb4-2:*lacZ* (**c**, n = 4 biologically independent animals) transgenic hearts collected two days after MI surgery show robust enhancer activity in ECs surrounding the injured tissue. Dashed grey lines indicate the injured myocardium, and boxed regions show magnified views within the injury (box 1), at the injury border zone (box 2), and at a remote region (box 3). Sections through age-matched control hearts are shown alongside, with comparable boxed regions selected for the magnified views. lv, left ventricle; rv, right ventricle. Grey scale bars represent 200 μm

expressing cells in the Dll4-12:*lacZ* and Ephb-4:*lacZ* transgenic hearts were found to co-express endothelial markers, while co-expression with cardiomyocyte markers was often observed (Supplementary Fig. 22). Similar patterns of expression were also seen seven days after injury with the Dll4in3mutMEF:*lacZ* and Notch1+16:*lacZ* transgenes (Supplementary Fig. 23). While this activity may originate from the *hsp68* minimal promoter within the transgene, the regionalised expression, lack of HLX-3:*lacZ* re-activation (which also contains *hsp68*) and previous accounts of low *hsp68*:*lacZ* transgene expression during cardiac regeneration make this unlikely[48]. These results may therefore indicate that the arterial and venous programmes are ectopically activated in ischaemic hearts. Although it is

possible that the *hsp68* minimal promoter may drive default cardiomyocyte expression in damaged tissues in the absence of a specific repressive or activating signal, the Dll4in3:*lacZ* transgene also showed only limited, cardiomyocyte-restricted activity after MI in the adult heart (Supplementary Fig. 23c). This enhancer is downstream of both angiogenic VEGFA-MEF2 and arterial RBPJ/SOXF programmes and, like HLX-3, can be reactivated in both arterial and angiogenic endothelium in other adult situations[19] (Supplementary Fig. 8e). This further demonstrates that the MEF2-angiogenic pathway is specifically repressed after MI regardless of other inputs. In conclusion, our results clearly demonstrate that neovascular growth in the ischaemic adult heart does not utilise the same pathways

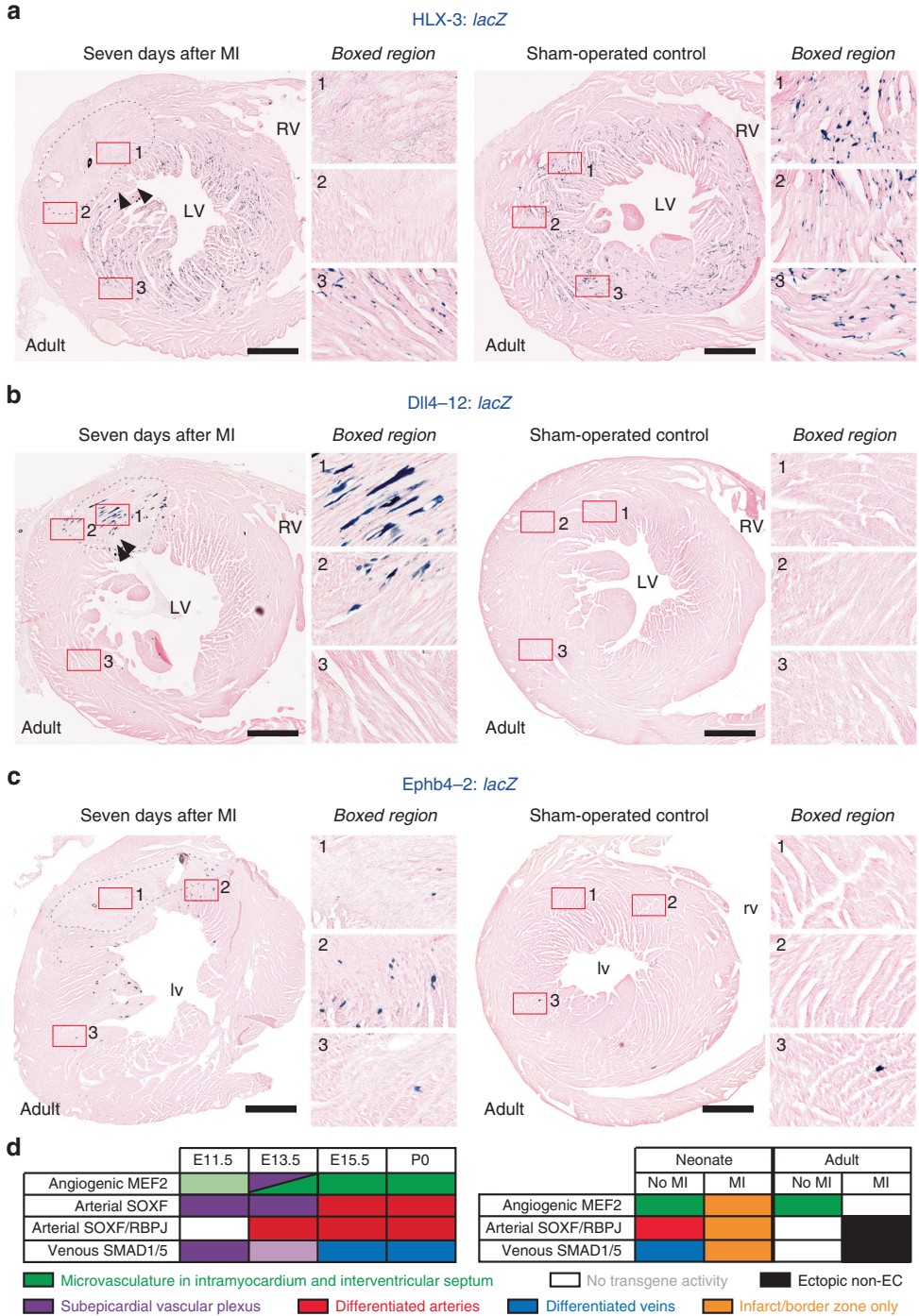

**Fig. 10** The adult heart does not activate developmental vascular regulatory pathways after MI. **a–c** Permanent ligation of the left anterior descending coronary artery was performed on transgenic adult male mice to induce MI. Representative sections through HLX-3:*lacZ* (**a**, *n* = 4 biologically independent animals), Dll4-12:*lacZ* (**b**, *n* = 5 biologically independent animals), and Ephb4-2:lacZ (**c**, *n* = 4 biologically independent animals) hearts collected 7 days after MI surgery compared to sham-operated controls. Additional time points after MI showed similar responses and can be seen in Supplementary Fig. 19, cumulative *n* values are *n* = 12 biologically independent animals for HLX-3:*lacZ*, *n* = 13 biologically independent animals for Dll4-12:*lacZ* and *n* = 12 biologically independent animals for Ephb4-2:*lacZ*. Infarct regions are indicated by grey dashed line, transgene expression is detected by X-gal staining (blue). Magnified views are shown of regions within the injury (box 1), at the injury border zone (box 2) and remote from the injury (box 3), and can be compared to sham-operated day seven controls. lv, left ventricle; rv, right ventricle. Black scale bars represent 500 µm. **d** Schematic summarising activity patterns of different vascular regulatory pathways in the coronary vasculature during development and after ischaemic injury

involved in developmental coronary vessel formation or in the damaged neonatal heart, and strongly indicates a fundamental divergence between the regulatory pathways intrinsically employed in the healthy and injured adult heart.

## Discussion

The presence of endothelial heterogeneity within the coronary vasculature is now generally accepted, although the extent and manner in which the different cellular sources contribute to

embryonic coronary vessels remain contested, and the regulatory mechanisms governing their growth are incompletely understood. Our results clearly demonstrate that the formation of the mature coronary vasculature requires numerous, independent signalling and transcriptional inputs. The data presented here is consistent with the ApjCreER/Nfatc1Cre lineage tracing-based model, in which complementary SV-derived and endocardial-derived endothelium unite to form the majority of the coronary vasculature[10,12]. However, our results indicate that the mechanisms used to generate these vessels may be different than previously suggested. Cre-based morphological analysis suggested that SV-derived ECs grow via sprouting angiogenesis, whereas endocardial-derived vessels use budding or compaction mechanisms[11,15,16,49]. However, the limited time-window of expression of the angiogenic enhancers in the early dorsal vascular plexus comparative to other regions suggests that sprouting angiogenesis is transient during SV-derived coronary vessel growth, whereas it remains the dominant regulatory pathway within endocardial-derived vessels throughout embryonic development and into post-natal growth. This strongly supports the idea, first postulated by Wu et al.[1], that endocardial-derived coronary vessels are formed using established angiogenic pathways downstream of VEGFA. This hypothesis is strengthened by the observation that the areas where endocardial sprouts emerge are heavily hypoxic[15,16]. Sprouting angiogenesis in the systemic vasculature occurs directly downstream of hypoxia-induced VEGFA secretion[50], so it is unsurprising that coronary vessels would respond to hypoxia via a similar regulatory pathway. Therefore it appears likely that the growth of new blood vessels in the systemic and coronary vasculature, although morphologically distinct, share a conserved regulatory pathway.

Unlike the angiogenic enhancers, both arterial and venous enhancers are robustly active in SV-derived coronaries. The punctate pattern of arterial enhancer activity in the E13.5 heart agrees with single-cell analysis identifying a pre-arterial population in the early SV-derived coronary sprout, while the steadily diminishing activity of venous enhancers in this region correlates with the known loss of venous identity in the coronary SV-derived plexus[11,34]. Therefore our results suggest that coronary arteriovenous differentiation utilises the same regulatory pathways as the systemic vasculature, despite disparate timing and cellular origins.

Although the enhancers utilised in this paper regulate expression of *Hlx*, *Dll4*, *Notch1*, *Ephb4* and *Coup-TFII/Nr2f2* genes in ECs, their activity patterns do not necessarily reflect the entire endogenous expression patterns of these genes. Instead, they specifically report the activity of the vascular VEGFA-MEF2, SOXF/NOTCH and BMP-SMAD1/5:SMAD4 regulatory pathways. However, while these regulatory pathways are active in only limited populations of ECs, the SOXF/RBPJ, MEF2 and SMAD families of transcription factors that activate them are expressed more widely[16,19,36]. The manner in which SOXF/RBPJ and SMAD1/5 are able to activate expression in specific subpopulations of coronary ECs is still not fully understood. Systemic venous regulation by SMAD1/5:SMAD4 is downstream of BMP4 through the ALK3/BMPR1A receptor, yet phosphorylated SMAD1/5 is also found in the coronary arteries[43]. Therefore it is likely that additional direct transcriptional regulators of these enhancers, yet to be identified, also play a role in venous coronary specification and development. This may explain the discrepancies between the systemic and coronary venous response to loss of *Smad4*, as deletion of *Smad4* after E10.5 has a profoundly deleterious effect on systemic venous formation while coronary veins are reported to be relatively normal[22,43]. However, these two observations were made independently and overall lethality also varied, suggesting that differences in mouse background,

Cre-driver and tamoxifen administration may have influenced the outcome.

Systemic acquisition of arterial identity is thought to be directly downstream of VEGFA signalling[51], yet SV-derived coronary arteries form independently of this ligand[10]. However, both VEGFC and Elabela(ELA)/APJ signalling are required for SV-derived coronary formation[10], suggesting that they may interact with SOXF or RBPJ to activate gene expression within the heart and beyond. Intriguingly, VEGFC expression is also up-regulated in the arterial endothelium in the early embryo, while ELA/APJ signalling plays a crucial role in the formation of the dorsal aorta in zebrafish[52]. It will therefore be of great interest to establish how VEGFC or ELA/APJ signalling interacts with SOXF during the regulation of endothelial identity in both the systemic and coronary vasculature.

The regulation of MEF2 transcriptional activity is already well studied. In particular, MEF2 factors are directly bound and repressed by class II HDACs[53]. In the systemic vasculature, VEGFA-induced class II HDAC phosphorylation and subsequent shuttling to the cytoplasm allows the recruitment of EP300, and consequently the transcriptional activation of MEF2 during sprouting angiogenesis[19,28,29]. Given the patterns of VEGFA expression in the heart, this mechanism may also be important for endocardial-derived coronary vessel growth. Counterintuitively, the SOXF factor SOX17 may also play a role in the regulation of MEF2-mediated transcriptional activation. SOXF factors can directly interact with MEF2 factors in ECs[54,55], and they have been implicated in angiogenic sprouting, in addition to arteriogenesis, in the systemic vasculature[56,57]. *Sox17* expression in the endocardial cells adjacent to compact myocardium is associated with an increased endocardial contribution to the coronary vasculature[16], suggesting that endocardial SOX17 may prime such cells to develop into coronary vessels in combination with MEF2. Other potential co-regulators of MEF2 transcriptional gene activation include Notch signalling, calcium-regulated protein kinases and MAP kinases[53,58,59]. While many of these components have already been studied in the myocardium, our findings have uncovered a previously unsuspected fundamental role for MEF2 in coronary vessel formation, and consequently provide a focus for further mechanistic studies.

Angiogenesis, by definition, simply means the formation of blood vessels from existing vasculature. Previous studies based on genetic lineage tracing clearly demonstrate that pre-existing ECs significantly contribute to neovascular growth in the heart after MI, clearly implicating some form of angiogenesis in this process. However, there have been multiple proposed sources and mechanisms of formation for these vessels, and the regulatory processes controlling this neovascularization has not been described[44,45,60–62]. Our results now clearly indicate a fundamental divergence between the regulatory pathways employed in the normal and injured adult heart, and strongly suggest that the developmental VEGFA-MEF2 sprouting angiogenesis pathway is not active in ischaemia-induced cardiac neovascularization.

Although the re-activation of the angiogenic MEF2-driven pathway in the ischaemic heart is an attractive option for regeneration, our results do not yet demonstrate the benefit of this approach over augmentation of the heart's endogenous neovascular response in this setting. It is also unclear whether the silencing of the MEF2-driven angiogenic pathway in the ischaemic adult heart represents a beneficial or pathological response. Although it has been presumed that re-activation of developmental pathways can assist in neovascularization in the adult[45], this has not been proven experimentally. Clinical trials using recombinant VEGFA to stimulate angiogenesis in patients with chronic myocardial ischaemia did not result in significant improvements in myocardial perfusion[63], a result potentially

explained by HDAC-mediated inhibition of MEF2 factors in these ischaemic regions: ischaemia has been shown to induce HDAC activity in the mouse heart after MI[64], and pharmacological HDAC inhibition after MI can promote neovascularization and cardiac repair[65]. However, any attempts to alter HDAC-MEF2 binding in the heart must consider the harmful roles of pathological MEF2 signalling in the myocardium, and the crucial role of HDACs in preventing this dysregulation[53].

Taken together, our results clearly demonstrate the limited benefits to be achieved by simply extrapolating observations made during healthy embryonic and neonatal coronary vessel growth, and strongly emphasise the vital importance of directly studying neovascular growth in the ischaemic adult heart.

## Methods

**Transgenic mouse lines**. All animal procedures comply with all relevant ethical regulations, were approved by the Clinical Medicine Local Ethical Review Committee, University of Oxford and licensed by the UK Home Office. The enhancer: lacZ transgenic mouse lines used in this study were as previously reported: Dll4in3: lacZ and Dll4-12:lacZ[7], Dll4in3mutMEF:lacZ and HLX-3:lacZ[19], NOTCH1+ 16: lacZ[21], CoupTFII-965:lacZ and EphB4-2:lacZ[22]. All transgenic mouse lines were originally established on a C57Bl10 × CBA/J background, and have been back-crossed at least five generations onto a pure C57Bl10 background. Both sexes were used for embryonic, neonatal and healthy adult samples (with adult defined as sexually mature mice over six weeks old), adult MI analysis was restricted to male mice and occurred only after ten weeks of age for welfare reasons. Dll4in3, Dll4-12, Coup-TFII-965 and EphB4-2 enhancers were from the mouse sequence, NOTCH1+ 16 and HLX-3 enhancers were from the human sequence. The Dll4in3 enhancer contains the same third intron sequence as the similar Dll4-F2 enhancer reported by[18], although it has an additional 100 bp of sequence on the 5′ end. As transgenic mice used were all from established lines no intra-line variation in transgene expression patterns was expected nor seen. The exception was HLX-3: lacZ activity in adult mice hearts, which showed intra- and inter-litter variation in intensity but no variation in expression pattern. This is likely associated with variations in angiogenesis rates in these hearts, and is represented in the relevant figures by representative pictures of each outcome. No experimental randomisation or blinding was used as this was not considered necessary.

HLX-3:tdTomato transgenic mice were made by first generating an NruI-mT-PolyA-XbaI construct using GeneArt Gene Synthesis (Invitrogen), based on the sequence for membrane-bound tdTomato (mT) reported for the Rosa26 mT/mG plasmid (Addgene, #17787). mT-PolyA was then cloned into the hsp68-lacZ Gateway construct to replace the lacZ reporter gene, the HLX-3 enhancer sequence cloned upstream of hsp68-mT using Gateway technology (Invitrogen), and linearised DNA microinjected into oocytes.

Genotyping was performed by polymerase chain reaction (PCR), with ear biopsies, tail tips or yolk sac samples digested overnight at 55 °C in 100 μl of GNT buffer (50 mM KCl, 1.5 mM MgCl$_2$, 10 mM Tris pH8.5, 0.01% gelatin, 0.45% Nonidet P40, 0.45% Tween 20) with 10 μl 10 mg/ml proteinase K (Fisher Scientific), and 2 μl used directly for the PCR reaction. In post-natal analysis of hearts, additional tissue was taken and X-gal stained to confirm genotype. In the rare event that lacZ-positive animals had no detectable staining in tissues away from the heart after MI, the sample was excluded from analysis.

**Whole-mount X-Gal staining**. β-galactosidase expression was detected by X-gal staining. Embryonic, neonatal or adult hearts were dissected in cold phosphate-buffered saline (PBS) and fixed in 2% paraformaldehyde (PFA), 0.2% glutaraldehyde in PBS at 4 °C for 30 min to 2 h, depending on age, followed by two washes in Rinse solution (2 mM MgCl$_2$, 0.2% Nonidet P40, 0.1% sodium deoxycholate in PBS). They were then incubated in Staining solution (Rinse solution containing 1 mg/ml 5-bromo-4-chloro-3-indolyl β-d-galactopyranoside (X-gal), 5 mM K$_4$Fe(CN)$_6$ and 5 mM K$_3$Fe(CN)$_6$) at room temperature for 3 h to overnight, depending on staining intensity and age of sample. Imaging of whole hearts was performed using a stereo microscope (Leica M165C) equipped with a ProgRes CF Scan camera and ProgRes CapturePro software (Jenoptik).

For samples at embryonic day (E)13 and younger, following whole-mount imaging the X-gal-stained hearts were dehydrated through an ethanol series, cleared using Histo-Clear (National Diagnostics) and paraffin-embedded for sectioning. Sections measuring 10 μm were de-waxed, counter-stained using Nuclear Fast Red (Electron Microscopy Services) and imaged using an Axioplan 2 upright microscope (Zeiss) with a ProgRes C5 camera and ProgRes CapturePro software (Jenoptik).

**X-Gal staining on cryosections**. Due to the limited penetration of the X-gal substrate in larger tissues, X-gal staining was also performed directly on cryosections for time points over E13.5, and for all adult and neonatal MI hearts. Dissected hearts were fixed for 1–3 h, depending on size, in 4% PFA in PBS at 4 °C, then incubated

overnight in 30% sucrose at 4 °C. They were then washed in a 50/50 mix of 30% sucrose/OCT Embedding Medium (Thermo Scientific), followed by a minimum of two washes in OCT before mounting over dry ice and storing at −80 °C.

Cryosections were cut at a thickness of 15 μm, and allowed to thaw at room temperature before washing in PBS to remove the OCT embedding medium. Sections were incubated in Fix solution (4% PFA, 2 mM MgCl$_2$, 5 mM EGTA in PBS) for 10 min at room temperature, before further washes in PBS. They were then incubated in Cryosection Staining solution (2 mM MgCl$_2$, 0.02% Nonidet P40, 0.01% sodium deoxycholate, 5 mM K$_4$Fe(CN)$_6$, 5 mM K$_3$Fe(CN)$_6$, and 1 mg/ml X-gal in PBS) in a humidified chamber at room temperature for 1 h to overnight, depending on staining intensity.

After staining, sections were washed in PBS, fixed in 4% PFA in PBS for 15 min, counter-stained with Nuclear Fast Red and imaged using a NanoZoomer S210 slide scanner with NDP.view2 viewing software (Hamamatsu).

**Immunostaining**. Cryosections were thawed at room temperature, washed in PBS and permeabilised in 0.5% Triton X-100 (Merck) in PBS for 10 min. They were then washed in PBS and incubated for 1 h in Blocking solution (10% donkey serum (Sigma-Aldrich), 4% foetal bovine serum (Sigma-Aldrich), 0.2% Triton X-100 in PBS) at room temperature. Sections were incubated overnight at 4 °C in primary antibodies diluted in the Blocking solution. After further PBS washes, sections were incubated for 1 h at room temperature with Alexa Fluor®-conjugated secondary antibodies (Life Technologies) diluted 1:500–1000 in Blocking solution. After final PBS washes, DAPI staining and mounting of slides using Fluoromount™ aqueous mounting medium (Sigma-Aldrich), confocal images were acquired with a Zeiss 710 MP confocal microscope and processed using Zen and ImageJ software.

Primary antibodies used were: rat anti-CD31 (1:200, Dianova DIA-310), armenian hamster anti-CD31 (1:100, Abcam ab119341), biotinylated Isolectin B4 (1:200, Vector Laboratories B-1205), rat anti-Endomucin (1:50, Santa Cruz sc-65495), goat anti-DLL4 (1:50, R&D Systems AF1389), goat anti-EphB4 (1:50, R&D Systems AF446), mouse anti-Actin, α-Smooth Muscle—Cy3™ (1:200, Sigma-Aldrich C6198), mouse anti-Actin, α-Smooth Muscle (1:300, Sigma-Aldrich A5228), goat anti-SOX17 (1:300, R&D Systems AF1924), rabbit anti-Cardiac Troponin I (1:100, Abcam ab47003), rabbit anti-RFP (1:1000, Rockland 600-401-379, used to detect tdTomato), chicken anti-β-galactosidase (1:500, Abcam ab9361), rabbit anti-MEF2A (1:100, Abcam ab109420), rabbit anti-MEF2C (1:1000, Cell Signalling 5030S) and rabbit anti-MEF2D (1:1000, Abcam ab32845).

To compare β-galactosidase expression with various molecular markers, X-gal staining was performed immediately following this immunostaining protocol. After confocal imaging, slides were soaked in MilliQ water overnight at 4 °C, cover slips removed and the X-gal staining protocol performed. In some cases, a short X-gal stain was instead performed prior to the immunostaining protocol, with staining stopped as soon as colour started to develop. X-gal-stained sections were fixed in 4% PFA in PBS for 10 min, washed in PBS and then the immunostaining protocol proceeded as above. Images showing overlaps of X-gal staining with immunohistochemistry using this method are therefore digitally pseudo-aligned, as the images are taken sequentially at least two days apart and some subtle tissue movement (e.g., dehydration or swelling) was experienced.

**Adult MI model using permanent LAD ligation**. Male transgenic mice aged 10 weeks or older were anaesthetised with isoflurane (2% vol/vol in O$_2$) under external ventilation (~200 strokes min$^{-1}$; stroke volume ~200 μl min$^{-1}$) through the insertion of an endotracheal tube and underwent a thoracotomy. MI was induced by permanent ligation of the LAD with an 8.0 Ethilon suture, using aseptic technique. Sham-operated animals underwent a thoracotomy and then the suture passed through the equivalent position of the left ventricle but not ligated. Buprenorphine was delivered as a 0.10 mg ml$^{-1}$ solution via subcutaneous injection 10 min before the procedure to provide analgesia.

Sample size was at least four animals with detectable infarct region/time-point analysed/transgenic line. Power calculations were not used to estimate sample size as the output was binary (enhancer on/off around infarct). Hearts with no detectable infarct after surgery were excluded from analysis. No experimental randomisation or blinding was used as this was not considered necessary and identity of transgenic line was obvious by control staining patterns.

**Neonatal LAD ligation**. Pups were anaesthetised by hypothermia and isoflurane (4% vol/vol in O$_2$) and placed on ice until motionless, with reduced heartbeat and respiration. After creating an incision between two ribs on the left side, the heart was extruded and the left anterior descending artery was ligated with a 8.0 Ethilon suture. The heart was repositioned within the chest, the ribs sewn together and the skin sutured (7.0 Prolene suture). The pup was removed from the ice pack and recovered by rapid warming using a heat lamp, with administration of oxygen to aid recovery. Once regular breathing and independent movement were restored, the pup was placed in a recovery chamber with the rest of the litter and mother.

Hearts were dissected two, seven and 14 days post-permanent LAD ligation, fixed in 4% PFA in PBS for 2 h at room temperature, equilibrated overnight in 30% sucrose/PBS and embedded in OCT for cryosectioning).

Sample size was at least four animals with detectable infarct region/transgenic line. Power calculations were not used to estimate sample size as the output was

binary (enhancer on/off around infarct). Hearts with no detectable infarct after surgery were excluded from analysis. No experimental randomisation or blinding was used as this was not considered necessary and identity of transgenic line was obvious by control staining patterns.

**Assays of adult angiogenesis**. For Matrigel assays, HLX-3:lacZ transgenic mice were injected subcutaneously in the flank with BD Matrigel basement membrane matrix (BD) supplemented with 2 μg/mL fibroblast growth factor (Peprotech). Matrigel plugs were harvested 14 days after injection and X-gal stained using the same protocol as E13.5 embryos. For tumours, transgenic mice were subcutaneously injected with 100 μL of BD Matrigel basement membrane matrix (BD) containing $1 \times 10^5$ B16F10 melanoma cells. Tumours were harvested at approximately 12-mm diameter and X-gal stained using the same protocol as E13.5 embryos. The adVEGFA angiogenesis assay provides a way to temporally investigate pathological angiogenesis[66]. Mice were injected intradermally on the dorsal side of each ear with 10 μl of adVEGFA diluted 1:30 in 3% glycerol/PBS. At the required time-point ears were harvested and skin removed from the dorsal side. Ears were fixed in 2% paraformaldehyde 0.2% glutaraldehyde in PBS for 20 min at 4°, washed twice in PBS then X-gal stained overnight at room temperature. Ears were then placed in 4% paraformaldehyde for storage.

**Reporting summary**. Further information on research design is available in the Nature Research Reporting Summary linked to this article.

## Data availability
The authors declare that the main data supporting the findings of this study are available within the article, its Supplementary Figures and Methods. All additional data that supports the findings of this study are available from the corresponding author upon request.

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

## Acknowledgements

Funding was provided by the British Heart Foundation (PG/10/83/28610 and PG/16/34/32135 to N.S., S.P. and S.D.; PG/16/27/32114; to A.N.R. and N.S.), by the BHF Oxbridge Centre of Regenerative Medicine (RM/13/3/30159; to N.S.) and by the BBSRC (BB/L02038/1; to S.D. and A.N.). We thank M. Shipman and R. Lisle for help with imaging. This work was also supported by a British Heart Foundation Ian Fleming Senior Research Fellowship (FS/13/4/30045; N.S.), a British Heart Foundation Senior Research Fellowship (FS/1735/32929; S.D. and A.N.), and by the Ludwig Cancer Research Ltd. (S.D)

## Author contributions

Conceptualisation: S.D.; Methodology: S.P., N.S. and S.D; Investigation, S.P., M.G-R., A.N., A.N.R., J.P., K.C., I.R., N.S. and S.D.; Resources: N.S. and S.D.; Writing—Original Draft, S.D.; Writing—Review and Editing, S.P., N.S and S.D.; Visualisation: S.P. and S.D.; Supervision: N.S and S.D., Project administration: S.D.; Funding acquisition: N.S. and S.D.

## Additional information

**Competing interests:** The authors declare no competing interests.

