## [Peer Review File · Nature Communications]

Reviewers' Comments:

Reviewer #1:

Remarks to the Author:

In this study, the authors describe the activity of well-characterized endothelial enhancers in the developing mouse heart, identifying two independent coronary vessel regulatory pathways: The VEGFA-MEF2 angiogenic pathway was predominantly active in endocardial derived coronary vessels during embryonic development, whereas a second VEGFA-MEF2 dependent enhancer Dll4in3 (SOXF/RBPJ arterial pathway) was specifically active in differentiated arterial coronaries. The authors also show that while both pathways contributed to post-natal coronary vascularization and post-MI neovascular growth in the neonate, the angiogenesis-associated VEGFA-MEF2 pathway was unexpectedly repressed in the adult heart after MI while the arterial Dll4-12 enhancer was activated after MI, demonstrating a fundamental divergence between the regulation of coronary vessel growth in healthy and ischemic adult hearts.

The data are in general of good quality and are well presented, the main caveat of this MS is that the results are very descriptive, and no functional analysis besides using mutated version of the enhancers was performed.

Reviewer #2:

Remarks to the Author:

In Payne et al., the author's use a collection of well-characterized vascular enhancer reporters to thoroughly characterize the signaling pathways activity during coronary vessel development and cardiac injury. The expression patterns of these enhancers brings to light important aspects in coronary development and demonstrates that embryonic pathways are not normally active in the adult following myocardial infarction. They also implicate upstream signaling molecules. These use of these unique tools are extremely useful and relevant for understanding this complex vascular bed, elements of which remain controversial. Specifically, the study provides evidence that the endocardium is primarily activated by VEGF-A-MEF2, while the SV less so, and that this activation cannot be normally stimulated in the non-regenerative adult heart. Also, the study uses an arterial-specific enhancer to suggest a time point when the arterial program is initiated, which is another pathway not activated in the coronaries of the injured adult. The enhancers suggest that for the SV a SOX activated pathway precedes a SOX/RBPJ combinatorial activation for full arterial differentiation. In general, the enhancers provide compellingly information on how coronary vessels from the different cell types respond differently. The major unresolved issue in the manuscript is that the authors need to provide the reader with more views of the expression patterns and more carefully evaluate their conclusions regarding the two progenitor sources.

Major comments:

1. The authors need to more clearly point out the nature of transgenic enhancer reporters. For example, when reporters are placed within the gene, all cells expressing the gene are generally positive. In contrast, recombination efficiency makes Cre-induced reporter labeling an underestimation of the number of positive cells. Which scenario are the authors reporter's more like and what does this mean for the results?

2. Because many of the conclusions rely on the fact that the SV and endocardium produce vessels in different areas, the authors should more thoroughly demonstrate the regionalization of the enhancers and pay more attention to showing subepicardial versus intramyocardial. For example, whole heart transverse sections at multiple levels with adjacent sections stained for endothelial cells and, preferably also endocardial cells (i.e. Emcn), would more clearly demonstrate exactly where the signal in the whole mounts is located. The high magnification section images are not that helpful because we do not know where exactly they are located. This applies to all

developmental figures and is my major critique of the conclusions in this study.

3. The authors should be careful about some of their conclusions on SV versus endocardium if they are unable to simultaneously include lineage tracing. The dorsal subepicardial surface and ventricular septum is easily distinguishable and definitive conclusions can be made. But in the myocardium, the SV and endocardial-derived vessels do some mixing:

What is very clear—

- VEGFA-MEF2 activity is high in the septum and is exclusively in the postnatal 2nd CVP that arises from the endocardium after birth.
- VEGFA-MEF2 is absent from initial e11.5 SV vessels while Notch+16 is on. This is important for understanding the initiation of SV angiogenesis. I would recommend showing this comparison in the main figure. Since the SV is a vein, is it possible that BMPs (Wiley et al., 2011, NCB) are the activating signal? Do the authors have a BMP enhancer?

What is less clear—

- Specificity of Dll4-12 arterial enhancer to SV (some coronary vessels from endocardial cells in the mixing intramyocardial region could activate this enhancer). The authors could do side-by-side comparisons (tissue sections) with VEGFA-MEF2 and Dll4-12 and show that they are in exclusively different domains, i.e. inner vs outer myocardium.
- Specificity of MEF enhancers to endocardium, particularly in the intramyocardium (SV vessels may or may not respond to VEGFA when they make it to the myocardium). The analysis suggested directly above could also help with this point.
- Specificity of Notch1+16. It appears to have some endocardial-derived signal in Fig. 3F e13.5 and G e15.5 ventral face.

4. The readability and interpretation of the results with regard to SV versus endocardium could be much improved with some schematics of where the respective vessels are located. Also, more arrows on figures showing the same regions on their whole mounts and tissue sections. Throughout the manuscript figures are cited, but we are not exactly sure where to look for the result they are explaining.

5. Figure 1D. Dll4 cannot be used as a differentiated artery marker as it is also expressed in angiogenic vessels as shown in the authors previous work. This analysis needs others such as SMA, etc.

6. Is Dll4in3 and Dll4-12 off in adult arteries? This is an important finding and could be focused on and discussed.

7. The authors need to provide a model summarizing their findings.

8. Is it practical in this study to answer why the enhancers are not active in adults? Are VEGFA/MEF/SOX not expressed or is the enhancer chromatin not accessible? It seems like ISH, qPCR and/or ChIP experiments could at least begin to address this issue.

Minor points:

9. What is the valve cushion activity seen in Hlx? Why is this not in the Dll4in3 enhancer?

10. Figure 3d is not cited

11. Is the artery pathway ever seen in endocardial cells themselves prior to their incorporation into coronary vessels?

Reviewer #3:

Remarks to the Author:

In this manuscript, the authors analyzed expression patterns of four representative endothelial cell enhancers in coronary vessel development and post myocardial infarction (MI) in neonatal and adult hearts. By observing the enhancer activities, the authors deduced mechanisms of coronary vessel development derived from endocardium vs sinus venosus (SV) that are consistent with previously reported results using Cre-Lox based lineage tracing but also added new perspectives. For instance, they demonstrated that VEGFA-MEF2 pathway was mainly active in endocardial-derived angiogenic coronary endothelial cells, whereas SOXF/RBPJ was active in coronary artery and SV derived vessels. These enhancer activities persisted to neonatal or even adult stage and were reactivated in neonatal mouse MI models. However, the enhancers are not re-activated in adult hearts after MI. Although this manuscript provides interesting findings for coronary vessel development and revascularization, it is very descriptive with not many mechanical insights. I have a few major concerns that dampened my enthusiasm. Overall, there is a disconnect between the developmental studies with the neovascularization after MI.

1. How do enhancers reflect endogenous gene expression during these processes? Furthermore, enhancer activity does not always represent function. Is endogenous MEF2 expressed in coronary endothelial cells? If so, is it expressed in endocardium vs SV derived coronary endothelial cells? It will be important to determine the endogenous MEF2 expression in coronary vessel in order to interpret these data. Furthermore, are MEF2 KO mice shown to have coronary vessel phenotypes?

2. The authors only looked at 4 representative enhancers. It is not convicting to generalize that developmental processes of coronary vessel formation are not reactivated in neovascularization during injury and repair. Furthermore, there might be multiple enhancers for the same genes. I am not clear how the authors exclude the possibilities that different developmental enhancers of the same genes are activated during revascularization. It is also not clear whether the authors proposed that these enhancers were actively repressed during neovascularization after heart injury or simply not utilized/activated.

3. The orientation and labels of Figure 1E suggest that HLX-3 activity is restricted in the right ventricle. However, the sections shown in Figure 1F suggest that the enhancer activity is in the LV. These data are not consistent.

4. The adult hearts are much bigger than the embryonic hearts. Are the sections images representative of different levels/regions of the hearts?

5. Some of the samples sizes are very small for the neonatal and adult heart studies (n=2).

Reviewers' comments:

Reviewer #1 (Remarks to the Author):

*In this study, the authors describe the activity of well-characterized endothelial enhancers in the developing mouse heart, identifying two independent coronary vessel regulatory pathways: The VEGFA-MEF2 angiogenic pathway was predominantly active in endocardial derived coronary vessels during embryonic development, whereas a second VEGFA-MEF2 dependent enhancer *Dll4in3* (SOXF/RBPJ arterial pathway) was specifically active in differentiated arterial coronaries. The authors also show that while both pathways contributed to post-natal coronary vascularization and post-MI neovascular growth in the neonate, the angiogenesis-associated VEGFA-MEF2 pathway was unexpectedly repressed in the adult heart after MI while the arterial *Dll4-12* enhancer was activated after MI, demonstrating a fundamental divergence between the regulation of coronary vessel growth in healthy and ischemic adult hearts.*

The data are in general of good quality and are well presented, the main caveat of this MS is that the results are very descriptive, and no functional analysis besides using mutated version of the enhancers was performed.

We respectfully disagree with this reviewer's statement that enhancer:reporter expression analysis is purely descriptive. Unlike analysis of the expression of a single gene, the expression patterns of enhancer:reporter transgenes provide a wealth of detailed information about activity of signalling and transcriptional regulatory pathways that would be impossible to ascertain otherwise. Every vascular enhancer investigated in this study has already been characterized in detail, and the transcriptional and signalling pathways which activate its expression in endothelial cells have been carefully established and published in a peer-reviewed paper. In each case these studies includes functional analysis verifying that the regulatory pathway activating each enhancer is required for the development of the specific endothelial sub-type investigated. This data includes but is not limited to:

- functional analysis of decreased angiogenic sprouting in *Mef2a;Mef2c* compound endothelial null mice and VEGFA-mediated activation of MEF2 transcriptional activity (supporting the VEGFA-MEF2-driven angiogenic pathway analysis) ¹;
- functional analysis of loss of venous identity in *Smad4* and *Alk3* endothelial null mice and *smad1/5* morphant zebrafish (supporting the BMP-SMAD driven venous pathway analysis) ²;

- * functional analysis of loss of arterial identity and Dll4in3 activity in *soxF/rbpj* morphant and chemically repressed zebrafish and the loss of *Notch1* expression in *Sox7:Sox18* compound mutant mice (supporting the SOXF/RBPJ-driven arterial analysis)^{3,4,1}

Additionally, papers from other labs have also provided additional functional evidence for the involvement of these pathways in the endothelial sub-types specified. This data includes but is not limited to:

- demonstration that VEGFA activates Dll4in3 enhancer and *DLL4* in arteries⁵;
- evidence of loss of arterial identity in *Sox17* endothelial null mice⁶;
- *in vitro* evidence of VEGFA-induction of *HLX* expression⁷;
- predating our work on BMP-driven venous development, functional analysis demonstrating reduced venous sprouting in zebrafish after BMP inhibition or deletion⁸.

Therefore, while we appreciate that our chosen analytical tools are unusual, we strongly feel our observations are well supported by functional analysis. Further, such observations would be extremely difficult to acquire using other techniques. As we state in the Introduction, “precisely delineating the signalling and transcriptional cascades involved in coronary vessel growth and behaviour has been challenging: knockout models often suffer systemic vascular abnormalities prior to coronary vessel formation, while defects in one type of coronary vessel can be compensated by other sources⁹, confounding phenotypic analysis”. Consequently, we feel that these well-validated enhancer:reporters have provided us a fantastic opportunity to study the activity of multiple different vascular regulatory pathways in their endogenous setting (acting on endogenous enhancer elements instead of artificial response elements) in healthy wild-type mice.

Reviewer #2 (Remarks to the Author):

In Payne et al., the author’s use a collection of well-characterized vascular enhancer reporters to thoroughly characterize the signaling pathways activity during coronary vessel development and cardiac injury. The expression patterns of these enhancers bring to light important aspects in coronary development and demonstrates that embryonic pathways are not normally active in the adult following myocardial infarction. They also implicate upstream signaling molecules. The use of these unique tools are extremely useful and relevant for understanding this complex vascular bed, elements of which remain controversial. Specifically, the study provides evidence that the endocardium is primarily activated by VEGF-A-MEF2, while the SV less so, and that this activation cannot be normally stimulated in the non-regenerative adult heart. Also, the study uses an arterial-specific enhancer to suggest a time point when the arterial program is initiated, which is another pathway not activated in the coronaries of the injured adult. The enhancers suggest that for the SV a SOX activated pathway precedes a SOX/RBPJ combinatorial activation for full arterial differentiation. In general, the enhancers provide compellingly information on how coronary vessels from the different cell types respond differently. The major unresolved issue in the manuscript is that the authors need to provide the reader with more views of the expression patterns and more carefully evaluate their conclusions regarding the two progenitor sources.

Major comments:

1. The authors need to more clearly point out the nature of transgenic enhancer reporters. For example, when reporters are placed within the gene, all cells expressing the gene are generally positive. In contrast, recombination efficiency makes Cre-induced reporter labeling

an underestimation of the number of positive cells. Which scenario are the authors reporter's more like and what does this mean for the results?

This has now been included in the Introduction:

Once the precise upstream factors regulating a discrete endothelial enhancer have been defined, transgenic animal models expressing reporter genes under the control of these enhancers become powerful tools to study the roles of different regulatory pathways during vascular growth. This can provide more information than the expression pattern of a single regulatory pathway component: vascular regulatory factors are often expressed in wider domains than the genes they activate, requiring epigenetic modification and/or specific combinations with other factors for gene activation (e.g. ^{1,4}). Crucially, the use of endogenous enhancers also avoids complications associated with synthetic pathway response elements (e.g. the BMP and Notch response elements, ^{10,11}). Such elements are generated by multimerizing consensus binding motifs for a single transcriptional activator (e.g. SMAD motifs for BMP, RBPJ motifs for Notch), and therefore lack binding sites for both essential endothelial transcription factors (e.g. ETS motifs, found in all endogenous endothelial regulatory elements ¹²) or for co-factors required for context-specific activation. Consequently, response elements cannot reflect the endogenous vascular response to individual pathways. Enhancer:reporter transgenic models also differ from Cre-driven lineage tracing: reporter gene expression is not influenced by recombination efficiency and switches off once the enhancer is no longer active, thereby providing an accurate readout of when and where a particular regulatory pathway is activated. Enhancer:reporter mice are also distinct from transgenic reporters inserted within a given gene. In such cases, reporter expression is influenced by any number of regulatory factors acting through multiple (often undefined) enhancer and promoter interactions, whereas the expression of enhancer:reporters is driven by a limited number of factors interacting with the defined enhancer.

2. Because many of the conclusions rely on the fact that the SV and endocardium produce vessels in different areas, the authors should more thoroughly demonstrate the regionalization of the enhancers and pay more attention to showing subepicardial versus intramyocardial. For example, whole heart transverse sections at multiple levels with adjacent sections stained for endothelial cells and, preferably also endocardial cells (i.e. Emcn), would more clearly demonstrate exactly where the signal in the whole mounts is located. The high magnification section images are not that helpful because we do not know where exactly they are located. This applies to all developmental figures and is my major critique of the conclusions in this study.

We have now adjusted the Figures to include, for all enhancers, whole heart transverse sections of transgenic hearts stained with X-gal in combination with the required marker analysis. Where required, this analysis extends to multiple levels of each heart. For ease of visualization, X-gal staining is shown on the whole heart transverse section, while the overlap of this staining with endothelial markers is shown in accompanying higher magnification images of specific (clearly marked) regions within each whole heart section. Because the variable expression intensity of X-gal in each different enhancer line made consistent use of the β -gal antibody impossible, we instead used a two-stage process to enable us to compare X-gal staining directly with IHC for endothelial markers on the same section. This analysis, detailed in the Methods section, meant that the IHC analysis was conducted and imaged first, after which each section was X-gal stained and imaged again. These two images of a single section were then merged, with occasional slight changes in section size between the two images (caused by dehydration), which were compensated for in Photoshop.

3. The authors should be careful about some of their conclusions on SV versus endocardium if they are unable to simultaneously include lineage tracing. The dorsal subepicardial surface

and ventricular septum is easily distinguishable and definitive conclusions can be made. But in the myocardium, the SV and endocardial-derived vessels do some mixing:

What is very clear—

- VEGFA-MEF2 activity is high in the septum and is exclusively in the postnatal 2nd CVP that arises from the endocardium after birth.

- VEGFA-MEF2 is absent from initial e11.5 SV vessels while Notch+16 is on. This is important for understanding the initiation of SV angiogenesis. I would recommend showing this comparison in the main figure.

The analysis of Notch1+16 now constitutes Fig. 5.

Since the SV is a vein, is it possible that BMPs (Wiley et al., 2011, NCB) are the activating signal? Do the authors have a BMP enhancer?

We have recently published a paper describing two novel vein-specific enhancers, Ephb4-2 and CoupTFII-965. Both of these are directly bound and activated by SMAD1/5:SMAD4, downstream of BMP signalling². To investigate the role of this pathway in coronary vessel growth in development and regeneration, we have now expanded this paper to include analysis of these two enhancers. Analysis of the activity of BMP-driven venous enhancer:reporter transgenes in development can be found in Fig. 7 and Supplementary Figs. 15 and 16, while the behaviour of this pathway after ischemic injury has been included with the angiogenic and arterial pathways in Figs. 9 and 10, and Supplementary Fig. 19. Discussion of the behaviours of these enhancers during cardiac development is included in a new section in Results (“A BMP/SMAD-driven transcriptional pathway is active in SV-derived coronary venous endothelium:”), stating:

In agreement with the venous origin of SV-derived coronary sprouts, we saw robust activity of the Ephb4-2:lacZ transgene at the dorsal base of the heart at E11.5, the location where coronary vessels first emerge from the SV (Fig. 7b)¹³. However, transgene activity on this dorsal face was significantly diminished by E13.5, and from E15.5 activity was specifically detected only in a limited number of superficial coronary vessels primarily found on the dorsal aspect (Fig. 7b). This activity correlated with the venous markers EPHB4 and EMCN in superficial vessels, but did not overlap with EMCN-positive endocardium (Fig. 7c-d and Supplementary Fig. 15). No activity was seen in arterial cells, although limited transgene expression was seen in limited regions of the outflow tract (Fig. 7d). These results correlate with previous observations into the differentiation of SV-derived coronary vessels, which found that whilst these cells initially have a venous identity, this is lost at E12.5, before becoming reactivated in superficial vessels as they form the mature coronary veins¹⁴. As with the arterial enhancers, little Ephb4-2 activity was seen in adult tissues (Supplementary Fig. 15). Similar patterns of transgene activity were also observed in a second BMP-SMAD1/5:SMAD4 driven venous enhancer transgene, CoupTFII-965:lacZ, although this transgene (and endogenous COUPTF-II) was also expressed in both systemic and coronary lymphatic vessels²(Supplementary Fig. 16). These results therefore indicate a novel role for BMP signaling in the formation of the coronary vasculature, and suggest that a similar ALK3-BMP-SMAD1/5 pathway may regulate both systemic and coronary venous differentiation.

In ischemic hearts, the Ephb4-2:lacZ transgene was reactivated in neonatal endothelial cells, but only showed ectopic myocardial activity in adult hearts. This was similar to the arterial enhancers, and was discussed in combination with them in the relevant Results sections.

What is less clear—

- Specificity of Dll4-12 arterial enhancer to SV (some coronary vessels from endocardial

cells in the mixing intramyocardial region could activate this enhancer). The authors could do side-by-side comparisons (tissue sections) with VEGFA-MEF2 and Dll4-12 and show that they are in exclusively different domains, i.e. inner vs outer myocardium.

We have addressed this question in two different ways. Firstly, our greatly improved images of transverse sections from both Dll4-12:*lacZ* and HLX-3:*lacZ* embryos clearly demonstrate that these two transgenes are largely expressed in different domains of the heart by E15.5. However, this analysis did identify occasional HLX-3:*lacZ*-expressing endothelial cells in arterial positions. This is unsurprising, as early HLX-3:*lacZ* expressing embryos (E8-E11) show considerable transgene activity in arteries, although this is lost as they mature. Further, angiogenic endothelial cells are known to preferentially contribute to arteries in regenerating zebrafish fin and post-natal mice^{15,16}. Therefore, to further investigate their relative distribution, we also generated a novel line expressing a HLX-3:tdTomato transgene, in which the *lacZ* reporter gene is replaced with tdTomato, allowing us to cross it with the Dll4-12:*lacZ* line. Hearts from double transgenic mice are shown in Fig. 6 and Supplementary Fig. 14 and discussed in detail in a new section of the Results (“The angiogenic and arterial programmes are predominantly active in distinct coronary endothelial cells”) stating:

At E15.5, a timepoint when both pathways were robustly active within the heart (Dll4-12 in arterial-positioned vessels, HLX-3 restricted to a subset of less differentiated vessels), most coronary vessels showed activity of a single transgene, whilst others showed no transgene activity (Fig. 6 and Supplementary Fig. 14). However, we also found that a subset of coronary vessels, predominantly in arterial positions, expressed both transgenes (Fig. 6b-c). In approximately half of these cases, there was no overlap in transgene activity at the cellular level, with distinct endothelial cells within a single arterial cross-section expressing either the Dll4-12 or HLX-3 transgenes in a mosaic manner. In other such vessels with activity of both transgenes, we found some endothelial cells expressing both transgenes (Fig. 6c and Supplementary Fig. 14). These observations concur with our earlier separate analysis of HLX-3:*lacZ* and Dll4-12:*lacZ* lines in the heart, where we saw occasional arterial endothelial cells expressing HLX-3:*lacZ* or lacking Dll4-12:*lacZ* (Supplementary Fig. 4, 10-12). Due to the half-life of the reporter genes, we were unable to establish whether these represent a single endothelial cell in which both the SOXF/RBPJ and VEGFA-MEF2 pathways are active, or whether these cells are transitioning from angiogenic to arterial identity, or vice versa. Previous reports suggest the former may be more likely, as angiogenic cells are known to contribute to arterial identity in the systemic vasculature, whereas there is no evidence of the reverse^{2,15,16}. These results also align with observations from Cre-driven lineage tracing: both the SV-associated ApjCreER and endocardial-associated Nfatc1Cre labelled coronary vessels on the dorsal side of E15.5 hearts, with the cumulative percentage of cells labelled by either pathway significantly above 100%¹³. In conclusion, these results demonstrate that while the RBPJ/SOXF and VEGFA-MEF2 pathways are predominantly active in distinct regions of the coronary vasculature, a single vessel can contain endothelial cells responding to multiple different regulatory inputs.

• *Specificity of MEF enhancers to endocardium, particularly in the intramyocardium (SV vessels may or may not respond to VEGFA when they make it to the myocardium). The analysis suggested directly above could also help with this point.*

These points have been addressed by the remade figures and double transgenic mouse analysis discussed above.

• *Specificity of Notch1+16. It appears to have some endocardial-derived signal in Fig. 3F e13.5 and G e15.5 ventral face.*

We hope the additional transverse sections added to Fig. 5 and Supplemental Fig. 13 have addressed these concerns. Although we agree that the staining on the ventral face did

indicate possible expression in endocardial-derived regions, analysis of these sections suggest that there is very little, if any, transgene expression in endocardial-derived regions. In particular, there was no staining detected in the septum.

4. The readability and interpretation of the results with regard to SV versus endocardium could be much improved with some schematics of where the respective vessels are located. Also, more arrows on figures showing the same regions on their whole mounts and tissue sections. Throughout the manuscript figures are cited, but we are not exactly sure where to look for the result they are explaining.

A schematic has been provided in Fig. 1c. Arrows have been added to all Figures and citations of figures have been improved.

5. Figure 1D. Dll4 cannot be used as a differentiated artery marker as it is also expressed in angiogenic vessels as shown in the authors previous work. This analysis needs others such as SMA, etc.

We have now used a combination of DLL4, α -SMA and SOX17, and the exclusion of EMCN, as markers of arteries.

6. Is Dll4in3 and Dll4-12 off in adult arteries? This is an important finding and could be focused on and discussed.

Loss of Dll4in3:*lacZ* and Dll4-12:*lacZ* activity in mature arteries is now shown in Supplementary Figs. 8 and 11, and is referred to briefly in the Results. However, we did not consider this particularly relevant, as it could be reactivated in adult tissues when stimulated (as seen in ¹ and Supplementary Fig. 8e) and therefore likely indicated the quiescent state of most mature arteries.

7. The authors need to provide a model summarizing their findings.

A model is now included as Fig. 10d.

8. Is it practical in this study to answer why the enhancers are not active in adults? Are VEGFA/MEF/SOX not expressed or is the enhancer chromatin not accessible? It seems like ISH, qPCR and/or ChIP experiments could at least begin to address this issue.

We include analysis demonstrating the MEF2 factors themselves are expressed in ischemic neovascularization, indicating a potential role for epigenetics in this repression. This is discussed extensively in Discussion:

The epigenetic regulation of MEF2 transcriptional activity is already well studied. In particular, MEF2 factors are directly bound and repressed by class II HDACs¹⁷. In the systemic vasculature, VEGFA-induced class II HDAC phosphorylation and subsequent shuttling to the cytoplasm allows the recruitment of EP300, and consequently the transcriptional activation of MEF2 during sprouting angiogenesis^{1,18,19}. Since MEF2 factors are expressed throughout the coronary vasculature, it is likely that epigenetic regulation. Given the patterns of hypoxia and VEGFA expression in the heart, this mechanism may also be important for endocardial-derived coronary vessel growth. Counterintuitively, the SOXF factor SOX17 may also play a role in the regulation of MEF2-mediated transcriptional activation. SOXF factors can directly interact with MEF2 factors in endothelial cells^{20,21}, and have been implicated in angiogenic sprouting, in addition to arteriogenesis, in the systemic vasculature^{22,23}. Sox17 expression in the endocardial cells adjacent to compact myocardium is associated with an increased endocardial contribution to the coronary vasculature⁹, suggesting that endocardial SOX17 may prime such cells to develop into coronary vessels in combination with MEF2. Other potential co-regulators of MEF2 transcriptional gene

activation include Notch signalling, calcium-regulated protein kinases and MAP kinases^{17,24,25}. While many of these components have already been studied in the myocardium, our findings have uncovered a previously unsuspected fundamental role for MEF2 in coronary vessel formation, and consequently provide a novel focus for further mechanistic studies.

We agree with the reviewer that analysis of the mechanisms repressing these enhancers in ischemic adult hearts is very interesting, but strongly feel it is beyond the scope of this paper. We have been recently awarded a three year grant focused just on repression of the MEF2-pathway in ischemic hearts, which is indicative of the extensive research required to understand the mechanisms behind these observations. This is in part because MEF2 factors have multiple co-regulators, most of which have already been implicated in aspects of cardiac development and regeneration. Decoupling these will require detailed analysis.

Minor points:

9. *What is the valve cushion activity seen in Hlx? Why is this not in the Dll4in3 enhancer?*

There is some expression of HLX-3:lacZ in the valve cushion that is not seen in any other transgene, including Dll4in3:LacZ. This staining can be seen clearly in Figure 1 and associated Supplementary Figures. The lack of Dll4in3 activity in valve cushion relative to HLX-3 is now briefly discussed in the Results section:

Interestingly, no activity was seen in the valve cushions, suggesting that the HLX-3 valve cushion activity was either a secondary consequence of transgene insertion, or downstream of non-MEF2 factors activating HLX-3 but not Dll4in3.

10. *Figure 3d is not cited*

This has been addressed.

11. *Is the artery pathway ever seen in endocardial cells themselves prior to their incorporation into coronary vessels?*

We have added images of E10 hearts from HLX-3:lacZ, Dll4in3:lacZ and Dll4-12:lacZ transgenic mice. In each case, activity is seen in the endothelium lining the outflow tract for all three, reflecting their shared activity in arterial locations at this time-point^{1,4}. Dll4-12 is never active in any other regions of the endocardium. However, both HLX-3 and Dll4in3 drive transient and fairly weak expression in the endocardium at this time-point, but by E13.5 very little expression is seen in the endocardium and none is detected at E15.5.

--

Reviewer #3 (Remarks to the Author):

In this manuscript, the authors analyzed expression patterns of four representative endothelial cell enhancers in coronary vessel development and post myocardial infarction (MI) in neonatal and adult hearts. By observing the enhancer activities, the authors deduced mechanisms of coronary vessel development derived from endocardium vs sinus venosus (SV) that are consistent with previously reported results using Cre-Lox based lineage tracing but also added new perspectives. For instance, they demonstrated that VEGFA-MEF2 pathway was mainly active in endocardial-derived angiogenic coronary endothelial cells, whereas SOXF/RBPJ was active in coronary artery and SV derived vessels. These enhancer activities persisted to neonatal or even adult stage and were reactivated in neonatal mouse MI models. However, the enhancers are not re-activated in adult hearts after MI. Although this manuscript provides interesting findings for coronary vessel development and revascularization, it is very descriptive with not many mechanical insights. I have a few major concerns that dampened

my enthusiasm. Overall, there is a disconnect between the developmental studies with the neovascularization after MI.

1a. How do enhancers reflect endogenous gene expression during these processes? Furthermore, enhancer activity does not always represent function. Is endogenous MEF2 expressed in coronary endothelial cells? If so, is it expressed in endocardium vs SV derived coronary endothelial cells? It will be important to determine the endogenous MEF2 expression in coronary vessel in order to interpret these data.

We have now better explained the use of enhancers versus endogenous genes to investigate regulatory pathway activation in the Introduction, including:

Complex spatial and temporal patterns of gene transcription in mammals are primarily regulated by gene enhancers (also known as cis-regulatory elements). Enhancers are modular elements containing densely clustered groups of transcription factor binding motifs that work cooperatively to activate and enhance transcription²⁶. Our lab and others have recently characterized a number of enhancers which drive gene expression specifically to discrete sub-populations of endothelial cells. Analysis of these enhancers has clearly demonstrated the important roles of epigenetic modification and transcription factor combinations in achieving distinct patterns of gene expression in different parts of the endothelium^{1-5,27,28}.

Once the precise upstream factors regulating a discrete endothelial enhancer have been defined, transgenic animal models expressing reporter genes under the control of these enhancers become powerful tools to study the roles of different regulatory pathways during vascular growth. This can provide more information than the expression pattern of a single regulatory pathway component: vascular regulatory factors are often expressed in wider domains than the genes they activate, requiring epigenetic modification and/or specific combinations with other factors for gene activation (e.g.^{1,4}). Crucially, the use of endogenous enhancers also avoids complications associated with synthetic pathway response elements (e.g. the BMP and Notch response elements,^{10,11}). Such elements are generated by multimerizing consensus binding motifs for a single transcriptional activator (e.g. SMAD motifs for BMP, RBPJ motifs for Notch), and therefore lack binding sites for both essential endothelial transcription factors (e.g. ETS motifs, found in all endogenous endothelial regulatory elements,¹² or for co-factors required for context-specific activation. Consequently, response elements cannot reflect the endogenous vascular response to individual pathways. Enhancer:reporter transgenic models also differ from Cre-driven lineage tracing: reporter gene expression is not influenced by recombination efficiency and switches off once the enhancer is no longer active, thereby providing an accurate readout of when and where a particular regulatory pathway is activated. Enhancer:reporter mice are also distinct from transgenic reporters inserted within a given gene. In such cases, reporter expression is influenced by any number of regulatory factors acting through multiple (often undefined) enhancer and promoter interactions, whereas the expression of enhancer:reporters is driven by a limited number of factors interacting with the defined enhancer.

Regarding enhancer expression vs the endogenous protein they regulate (e.g. Dll4in3 expression versus endogenous *Dll4*): these enhancers largely recapitulate the expression patterns of their cognate endogenous gene within the vasculature, but not elsewhere (e.g. the Dll4in3 enhancer closely mimics the expression pattern of *Dll4* in endothelial cells, but not in neural tissue). This analysis was included in the original papers describing each enhancer, but we have also now also added immunohistochemistry comparing endogenous *Dll4* and *Ephb4* expression with the enhancers in the coronaries.

Regarding enhancer expression vs the transcription factors which bind them (e.g. HLX-3 vs MEF2 factors): as now clearly stated in the Introduction, the transcription factors involved in

regulation of genes in the vasculature often expressed in wider domains than the genes they activate, requiring epigenetic modification and/or specific combinations with other factors for gene activation. In the case of MEF2 factors, these are widely expressed in the coronary vasculature and cardiomyocytes, but are known to require additional epigenetic modifications to become transcriptional activators. We have now provided detailed information about MEF2 protein expression in the coronary vessels during development, in the adult and after ischemic injury in Supplementary Figs. 1, 2 and 21. We have also revised the Results to discuss this:

Within the systemic vasculature, the formation of new vessels from existing ones occurs primarily through sprouting angiogenesis downstream of a VEGFA-induced, MEF2-regulated transcriptional programme^{1,29}. During this process, VEGFA triggers the release of repressive HDACs bound to MEF2 factors. MEF2 transcription factors are expressed widely throughout the systemic endothelium, but the loss of HDAC epigenetically converts MEF2 factors from repressors to activators, allowing them to specifically activate enhancers during sprouting angiogenesis^{1,18,19,30,31}.

Conflicting models of coronary vessel formation have implicated sprouting angiogenesis exclusively to different compartments of the vasculature^{13,32,33}. Therefore, we wished to directly establish the role of the VEGFA-MEF2 sprouting angiogenic pathway in coronary vascular growth. As in the systemic vasculature, the MEF2 factors MEF2A, C and D were widely expressed throughout the embryonic and neonatal coronary endothelium (as well as in other cardiac cell types; Supplementary Fig. 1-2). However, expression of MEF2 factors does not indicate activity of the epigenetically modified VEGFA-MEF2 pathway. To determine this, we instead examined the cardiac activity of HLX-3:*LacZ* enhancer:reporter transgene, which is directly activated by the VEGFA-MEF2 pathway in endothelial cells: The HLX-3 enhancer, located 3kb upstream of the homeobox transcription factor *HLX*, activates gene expression specifically during sprouting angiogenesis in the systemic vasculature via direct and essential MEF2 binding¹ (Figure 1A)....

For ischemic hearts:

The absence of HLX-3:*lacZ* expression did not correlate with loss of MEF2 factor expression, as MEF2C was detected in the vasculature surrounding infarcts after MI in both neonatal and adult hearts (Supplementary Fig. 21). Since the VEGFA-MEF2 angiogenic pathway is strongly influenced by epigenetic modification of MEF2 factors¹, these observations suggest that repression of the ability of MEF2 factors to activate transcription may play a role in the loss of angiogenic HLX-3 activity in the adult ischemic heart.

1b. ...Furthermore, are MEF2 KO mice shown to have coronary vessel phenotypes?

Ablation of a single MEF2 factor in the vasculature has no known effect on embryonic or adult vasculature, but compound deletion of both MEF2A/MEF2C results in defective angiogenesis two days after deletion in the post-natal retina¹. No published studies have investigated the effects of triple MEF2A/C/D vascular deletion, although similar analysis in muscle uncovered a requirement for MEF2 factors in skeletal muscle regeneration that had been unclear from earlier single or compound null analysis³⁴.

MEF2 factors also have other crucial roles in the vasculature, and therefore absolute ablation of MEF2 factors is likely to have other vascular consequences that will make analysis of the direct role of MEF2 factors in the coronary vasculature impossible to interpret. Some of these issues are already discussed in our previous work on HLX-3 and MEF in angiogenesis¹. For example, MEF2 factors are required for the flow-mediated activation of *Klf2* and *Klf4*. Deletion of *Klf2* in the embryo prevents correct heart valve formation and results in mid-gestation lethality prior to coronary vascular formation³⁵. Further, compound *Klf2/4* endothelial deletion in adult results in rapid lethality due to loss of endothelial integrity³⁶. Endothelial MEF2 factors are also essential for the correct expression

of *Mmp10*, a secreted endoproteinase that degrades the vascular extracellular matrix ³⁷. Loss of MEF2-mediated repression of *Mmp10* also results in embryonic lethality prior to coronary vessel formation. It is partly for these reasons that we decided to investigate the role of MEF2 in coronary vessels with the approach used here. This is briefly mentioned in the Introduction:

However, precisely delineating the signalling and transcriptional cascades involved in coronary vessel growth and behaviour has been challenging: knockout models often suffer systemic vascular abnormalities prior to coronary vessel formation, while defects in one type of coronary vessel can be compensated by other sources ⁹, confounding phenotypic analysis.

2a. The authors only looked at 4 representative enhancers. It is not convicting to generalize that developmental processes of coronary vessel formation are not reactivated in neovascularization during injury and repair.

For the MEF2 angiogenic pathway, we analysed two different MEF2-driven enhancers (HLX-3 and Dll4in3) from different genes. These share no homology beyond MEF2 binding motifs and ETS binding motifs (found in all endothelial enhancers regardless of expression domain). Further, we also analysed Dll4in3mutMEF enhancer activity, which is identical to Dll4in3 except for the mutation in a single MEF2 motif, very clearly identifying a requirement for MEF2 factors for this activity in certain regions of the heart. For the SOXF arterial pathway, we analysed three different SOXF driven enhancers (Dll4-12, NOTCH+16, Dll4in3/Dll4in3mutMEF) driving three different genes. They shared no homology beyond SOXF, RBPJ (not NOTCH1+16) and ETS binding motifs. For the SMAD-driven pathway, we analysed two different SMAD1/5:SMAD4-driven enhancers (Ephb4-2 and CoupTFII-965) which shared no homology beyond SMAD1/5 and ETS binding motifs. Consequently, from a total of eight different enhancers, we feel able to conclude that the expression domains shared by each set of enhancers were likely a consequence of the specified transcription factors. Activity of these enhancers collectively correlates with the well-characterised contributions from sinus venosus and endocardium, which give rise to >90% of coronary endothelium. That these pathways are not reactivated in response to injury, in particular the angiogenic pathway proposed as the main mechanism for neovascularisation post-MI ³⁸, is a striking and important finding.

2b. Furthermore, there might be multiple enhancers for the same genes. I am not clear how the authors exclude the possibilities that different developmental enhancers of the same genes are activated during revascularization.

It is entirely probable that most of the genes included here have multiple enhancers, including enhancers not studied in this paper. However, this has no influence on the conclusions of our paper, as we study not the gene but the selected enhancers in isolation. Consequently, our conclusions are entirely based on the activity driven by a single enhancer, and therefore by the transcription factors, and upstream cognate signalling pathways, that directly bind only the specified enhancer. This is one of the advantages of our analysis: because we study the expression of reporter genes controlled by only the defined enhancer, we know the induction signals come through that exact enhancer, and not from elsewhere. This is now explained in the alterations to the introduction (see response to query 1a).

Regarding the revascularization part of this paper, it is entirely possible that different enhancers for the same genes may be involved in gene regulation in adult hearts after ischemic injury. What our results show is that these putative revascularization enhancers are unlikely to be activated by the angiogenic MEF2, arterial SOXF/RBPJ and venous SMAD regulatory pathways that direct coronary vessel formation during development.

2c. It is also not clear whether the authors proposed that these enhancers were actively repressed during neovascularization after heart injury or simply not utilized/activated.

It is not really possible to know if the enhancers are repressed in the ischemic adult heart, or just not activated (they are in some ways two sides of the same coin), and we have tried to make that clear in our results and discussion sections. However, the MEF2-driven HLX-3:lacZ and Dll4in3:lacZ transgenes are active in the healthy adult hearts but not active around the infarct although MEF2 factors are still present, suggesting some form of repression (e.g. HDAC factors repressing the ability of MEF2 factors to activate transcription) is a likely explanation.

3. The orientation and labels of Figure 1E suggest that HLX-3 activity is restricted in the right ventricle. However, the sections shown in Figure 1F suggest that the enhancer activity is in the LV. These data are not consistent.

We agree with the reviewer that these images were unclear. We have repeated these experiments and included two representative whole-mount images of adult HLX-3:lacZ transgenic mice accompanied by two whole heart transverse section images (Fig. 2a and d and Supplementary Fig. 6b). Transgene intensity varied between HLX-3:lacZ hearts in the adult but the pattern of transgene activity within the heart remained similar, as can be seen in these images.

4. The adult hearts are much bigger than the embryonic hearts. Are the sections images representative of different levels/regions of the hearts?

Scale bars have been included, and now all images have accompanying images of whole heart transverse sections with the regions included in zoomed images clearly marked. Where multiple images of the same hearts are shown, they are presented in base to apex order for clarity.

5. Some of the samples sizes are very small for the neonatal and adult heart studies (n=2).

This has been addressed. All experiments are repeated at least 4 times, and the adult MI experiments included at least 12 repeats over three time-points, all showing the same results.

Relevant references

1. Sacilotto, N. *et al.* MEF2 transcription factors are key regulators of sprouting angiogenesis. *Genes Dev* **30**, 2297–2309 (2016).
2. Neal, A. *et al.* Venous identity requires BMP signalling through ALK3. *Nat Commun* **10**, 453 (2019).
3. Chiang, I. K.-N. *et al.* SoxF factors induce Notch1 expression via direct transcriptional regulation during early arterial development. *Development* **144**, 2629–2639 (2017).
4. Sacilotto, N. *et al.* Analysis of Dll4 regulation reveals a combinatorial role for Sox and Notch in arterial development. **110**, 11893–11898 (2013).
5. Wythe, J. D. *et al.* ETS Factors Regulate Vegf-Dependent Arterial Specification. *Dev Cell* **26**, 45–58 (2013).
6. Corada, M. *et al.* Sox17 is indispensable for acquisition and maintenance of arterial identity. *Nat Commun* **4**, 2609 (2013).
7. Fish, J. E. *et al.* Dynamic regulation of VEGF-inducible genes by an ERK/ERG/p300 transcriptional network. *Development* **144**, 2428–2444 (2017).
8. Wiley, D. M. *et al.* Distinct signalling pathways regulate sprouting angiogenesis from the dorsal aorta and the axial vein. *Nat Cell Biol* **13**, 687–693 (2011).
9. Sharma, B. *et al.* Alternative Progenitor Cells Compensate to Rebuild the Coronary Vasculature in Elabela- and Apj-Deficient Hearts. *Dev Cell* (2017). doi:10.1016/j.devcel.2017.08.008
10. Monteiro, R. M. *et al.* Real time monitoring of BMP Smads transcriptional activity during mouse development. *Genesis* **46**, 335–346 (2008).
11. Souilhoul, C. *et al.* Nas transgenic mouse line allows visualization of Notch pathway activity in vivo. *Genesis* **44**, 277–286 (2006).
12. De Val, S. & Black, B. L. Transcriptional Control of Endothelial Cell Development. *Dev Cell* **16**, 180–195 (2009).
13. Chen, H. I. *et al.* The sinus venosus contributes to coronary vasculature through VEGFC-stimulated angiogenesis. *Development* **141**, 4500–4512 (2014).
14. Red-Horse, K., Ueno, H., Weissman, I. L. & Krasnow, M. A. Coronary arteries form by developmental reprogramming of venous cells. *Nature* **464**, 549–553 (2010).
15. Pitulescu, M. E. *et al.* Dll4 and Notch signalling couples sprouting angiogenesis and artery formation. *Nat Cell Biol* **19**, 915–927 (2017).
16. Xu, C. *et al.* Arteries are formed by vein-derived endothelial tip cells. *Nat Commun* **5**, 5758 (2014).
17. Potthoff, M. J. & Olson, E. N. MEF2: a central regulator of diverse developmental programs. *Development* **134**, 4131–4140 (2007).
18. Ha, C. H., Jhun, B. S., Kao, H.-Y. & Jin, Z.-G. VEGF stimulates HDAC7 phosphorylation and cytoplasmic accumulation modulating matrix metalloproteinase expression and angiogenesis. *Arterioscler Thromb Vasc Biol* **28**, 1782–1788 (2008).
19. Ha, C. H. *et al.* Protein Kinase D-dependent Phosphorylation and Nuclear Export of Histone Deacetylase 5 Mediates Vascular Endothelial Growth Factor-induced Gene Expression and Angiogenesis. *J Biol Chem* **283**, 14590–14599 (2008).
20. Hosking, B. M. *et al.* SOX18 directly interacts with MEF2C in endothelial cells. *Biochem Biophys Res Commun* **287**, 493–500 (2001).
21. Overman, J. *et al.* Pharmacological targeting of the transcription factor SOX18 delays breast cancer in mice. *eLife Sciences* **6**, (2017).
22. Lee, S.-H. *et al.* Notch pathway targets proangiogenic regulator Sox17 to restrict angiogenesis. *Circ Res* **115**, 215–226 (2014).
23. Kim, K. *et al.* SoxF Transcription Factors Are Positive Feedback Regulators of VEGF Signaling Novelty and Significance. *Circ Res* **119**, 839–852 (2016).
24. Shen, H. *et al.* The Notch coactivator, MAML1, functions as a novel coactivator for MEF2C-mediated transcription and is required for normal myogenesis. *Genes Dev* **20**, 675–688 (2006).

25. Ge, R. *et al.* Critical role of TRPC6 channels in VEGF-mediated angiogenesis. *Cancer Lett.* **283**, 43–51 (2009).
26. Maston, G. A., Evans, S. K. & Green, M. R. Transcriptional regulatory elements in the human genome. *Annu Rev Genomics Hum Genet* **7**, 29–59 (2006).
27. Becker, P. W. *et al.* An Intronic Flk1 Enhancer Directs Arterial-Specific Expression via RBPJ-Mediated Venous Repression. *Arterioscler Thromb Vasc Biol* **36**, 1209–1219 (2016).
28. Robinson, A. S. *et al.* An arterial-specific enhancer of the human endothelin converting enzyme 1 (ECE1) gene is synergistically activated by Sox17, FoxC2, and Etv2. *Dev Biol* **395**, 379–389 (2014).
29. Phng, L.-K. & Gerhardt, H. Angiogenesis: a team effort coordinated by notch. *Dev Cell* **16**, 196–208 (2009).
30. Lu, J., McKinsey, T. A., Zhang, C. L. & Olson, E. N. Regulation of skeletal myogenesis by association of the MEF2 transcription factor with class II histone deacetylases. *Mol Cell* **6**, 233–244 (2000).
31. Chan, J. K. L., Sun, L., Yang, X.-J., Zhu, G. & Wu, Z. Functional characterization of an amino-terminal region of HDAC4 that possesses MEF2 binding and transcriptional repressive activity. *J Biol Chem* **278**, 23515–23521 (2003).
32. Tian, X. *et al.* De novo formation of a distinct coronary vascular population in neonatal heart. *Science* **345**, 90–94 (2014).
33. Wu, B. *et al.* Endocardial cells form the coronary arteries by angiogenesis through myocardial-endocardial VEGF signaling. *Cell* **151**, 1083–1096 (2012).
34. Liu, N. *et al.* Requirement of MEF2A, C, and D for skeletal muscle regeneration. (2014).
35. Lee, J. S. *et al.* Klf2 is an essential regulator of vascular hemodynamic forces in vivo. *Dev Cell* **11**, 845–857 (2006).
36. Sangwung, P. *et al.* KLF2 and KLF4 control endothelial identity and vascular integrity. *JCI Insight* **2**, e91700 (2017).
37. Chang, S. *et al.* Histone deacetylase 7 maintains vascular integrity by repressing matrix metalloproteinase 10. *Cell* **126**, 321–334 (2006).
38. He, L. *et al.* Preexisting endothelial cells mediate cardiac neovascularization after injury. *J Clin Invest* (2017). doi:10.1172/JCI93868

Reviewers' Comments:

Reviewer #2:

Remarks to the Author:

The authors have addressed all my concerns. This manuscript is a very thorough analysis that provides important new information on the regulatory pathways activated/inactivated during coronary development and injury.

Reviewer #3:

Remarks to the Author:

The authors addressed most of my previous comments.

However, the following points are still not clear for previous comments "How do enhancers reflect endogenous gene expression" and "there might be multiple enhancers for the same gene".

Specifically for the new Figure 10 (the enhancer expression in adults after MO), do authors look at endogenous gene expression of HLX-3, Dll4 and Ephb4 by in situ hybridization to see if the transcripts of these genes are expressed in adult coronary vessels after MI? If this info was published, the authors could cite the references. To compare the endogenous gene expression and enhancer activities in adult hearts can clarify the conclusion the author made that "there is a fundamental divergence between the regulation of coronary vessel growth in healthy and ischemic adult hearts".

REVIEWERS' COMMENTS:

Reviewer #2 (Remarks to the Author):

The authors have addressed all my concerns. This manuscript is a very thorough analysis that provides important new information on the regulatory pathways activated/inactivated during coronary development and injury.

--

Reviewer #3 (Remarks to the Author):

The authors addressed most of my previous comments. However, the following points are still not clear for previous comments "How do enhancers reflect endogenous gene expression" and "there might be multiple enhancers for the same gene".

Specifically for the new Figure 10 (the enhancer expression in adults after MO), do authors look at endogenous gene expression of HLX-3, Dll4 and Ephb4 by in situ hybridization to see if the transcripts of these genes are expressed in adult coronary vessels after MI? If this info was published, the authors could cite the references. To compare the endogenous gene expression and enhancer activities in adult hearts can clarify the conclusion the author made that "there is a fundamental divergence between the regulation of coronary vessel growth in healthy and ischemic adult hearts".

We apologise that this is still not clear in the manuscript. These enhancers are not proxies for the genes they regulate (e.g. HLX-3:*lacZ* is not being used as a marker for *Hlx* expression) but instead act as proxies/markers for the transcriptional pathway that activates them (so activation of HLX-3:*lacZ* tells us where the VEGFA-MEF2 regulatory pathway is active in endothelial cells). It makes no difference to our conclusions if *Hlx* is regulated by a second enhancer which does switch on in the ischemic adult heart, as we are interested in the fact that the VEGFA-MEF2 regulatory pathway is silent, not *Hlx*. The same is true for Dll4-12:*lacZ* and the other enhancers – they are representative of their upstream regulatory pathways, not the expression of their nearest gene. We have made further adjustments to both Introduction and Discussion to reflect these facts:

Introduction, new (underlined) part is added to the fifth paragraph:

Once the precise upstream factors regulating a discrete endothelial enhancer have been defined, transgenic animal models expressing reporter genes under the control of these enhancers become powerful tools to study the behaviours of different regulatory pathways during vascular growth. These enhancer:reporter constructs are not proxies for the gene they regulate, as most gene loci contain multiple enhancers and a single enhancer is rarely active in the entire domain of the gene it regulates. Instead, these enhancer:reporter constructs provide an easily detected read-out of the activity of each different upstream regulatory pathway. This can provide more information than the expression pattern of a single regulatory pathway component.....

Discussion, new (underlined) part is added to the beginning of existing third paragraph):

Although the enhancers utilized in this paper regulate expression of *Hlx*, *Dll4*, *Notch1*, *Ephb4* and *Coup-TFII/Nr2f2* genes in ECs, their activity patterns do not necessarily reflect the entire endogenous expression patterns of these genes. Instead, they specifically report the activity of the vascular VEGFA-MEF2, SOXF/NOTCH and BMP-SMAD1/5:SMAD4 regulatory pathways. However, while these regulatory pathways are active in only limited populations of ECs, the SOXF/RBPJ, MEF2 and SMAD families of transcription factors that activate them are expressed more widely....